



# A multi-season investigation of glacier surface roughness lengths through in situ and remote observation

Noel Fitzpatrick[1], Valentina Radić[1], Brian Menounos[2]

[1] Department of Earth, Ocean, and Atmospheric Sciences, University of British Columbia, Vancouver, V6T 1Z4, Canada,

[2] Natural Resources and Environmental Studies Institute and Geography Program, University of Northern British Columbia, Prince George, V2N 4Z9, Canada

*Correspondence to*: Noel Fitzpatrick (nfitzpat@eoas.ubc.ca)

**Abstract.** The roughness length values for momentum, temperature, and water vapour are key inputs to the bulk

aerodynamic method for estimating turbulent heat flux. Measurements of site-specific roughness length are rare for glacier surfaces, and substantial uncertainty remains in the values and ratios commonly assumed when parameterising turbulence. Over three melt seasons, eddy covariance observations were implemented to derive the momentum and scalar roughness lengths at several locations on two mid-latitude mountain glaciers. In addition, two techniques were developed in this study for the remote estimation of momentum roughness length, utilising LiDAR-derived digital elevation models with a 1 x 1 m

resolution. Seasonal mean momentum roughness length values derived from eddy covariance observations at each location ranged from 0.7–4.5 mm for ice surfaces, and 0.5–2.4 mm for snow surfaces. From one season to the next, mean momentum roughness length values over ice remained relatively consistent at a given location (0–1 mm difference between seasonal mean values), while within a season, temporal variability in momentum roughness length over melting snow was found to be substantial (> an order of magnitude). The two remote techniques were able to differentiate between ice and snow cover, and

return momentum roughness lengths that were within 1–2 mm (<< an order of magnitude) of the *in situ* eddy covariance values. Changes in wind direction affected the magnitude of the momentum roughness length due to the anisotropic nature of features on a melting glacier surface. Persistence in downslope wind direction on the glacier surfaces, however, reduced the influence of this variability. Scalar roughness length values showed considerable variation (up to two and a half orders of magnitude) between locations and seasons, and no evidence of a constant ratio with momentum roughness length or each

other. Of the tested estimation methods, the Andreas (1987) surface renewal model returned scalar roughness lengths closest to those derived from eddy covariance observations. Combining this scalar method with the remote techniques developed here for estimating momentum roughness length may facilitate the distributed parameterisation of turbulent heat flux over glacier surfaces without *in situ* measurements.



## 1 Introduction

The turbulent fluxes of sensible and latent heat can form a major component of the surface energy balance (SEB) of a glacier, and substantially influence its rate of surface melt (Hock and Holmgren, 1996; Anderson *et al.*, 2010; Fitzpatrick *et al.*, 2017). With a lack of direct measurement on glaciers, the bulk aerodynamic method is commonly used to parameterise the turbulent fluxes, requiring input of roughness length values for momentum ($z_{0v}$), temperature ($z_{0t}$), and water vapour ($z_{0q}$). Observations of roughness length are rare on glacier surfaces, however. The majority of SEB studies use values and ratios from previous research on similar surface types (e.g. Gillet and Cullen, 2011; Giesen *et al.*, 2014), or treat roughness lengths as model tuning parameters (e.g. Braun and Hock, 2004; Sicart *et al.*, 2005), rather than obtaining site-specific measurements. This approach introduces uncertainty into turbulent flux estimation, as the transferability of roughness lengths between locations and seasons is unknown. Furthermore, parameterisation of the turbulent heat fluxes has been shown in previous studies to be highly sensitive to the implemented roughness lengths (up to a doubling of the calculated flux for one order of magnitude increase in $z_{0v}$), and to dominate over stability corrections as a source of uncertainty (Munro, 1989; Braithwaite, 1995; Brock *et al.*, 2000; Fitzpatrick *et al.*, 2017). The importance of accurate roughness length selection, as identified in these studies, highlights the need for further research on the spatial and temporal variability of their values on glacier surfaces, and on the methods used in their estimation.

The roughness length values are defined as the lower limits of integration in the bulk-gradient or 'K' theory parameterisation of the turbulent fluxes (Stull, 1988). $z_{0v}$ can be thought of as the height above the surface at which wind speed, extrapolated downwards along an assumed logarithmic profile, will reach its surface value. Similarly, $z_{0t}$ and $z_{0q}$ can be considered to be the heights at which temperature and specific humidity reach their surface values, respectively. $z_{0v}$ accounts for the effects of form drag on the near-surface wind profile due to the interaction of airflow with features on the surface. In many glacier studies and climate models (e.g. Van As, 2011; Fausto *et al.*, 2016), $z_{0v}$ values of 1 mm and 0.1.mm are used for ice and snow surfaces, respectively, and are often assumed constant with time. Where measurements have been obtained on glacier surfaces, however, a large range of $z_{0v}$ values have been recorded, with several orders of magnitude of variation between different glaciers and seasons (e.g. Van den Broeke *et al.*, 2005; Brock *et al.*, 2010). In addition, existing values for $z_{0v}$ on glaciers (observed through mast-based vertical wind profile measurements, or estimated from eddy covariance (EC) observations or microtopography surveys) only provided values for an individual location or turbulent footprint. Implementing these single values in a glacier-wide distributed model or in a point model at another location on the glacier may not account for the potential variability in surface roughness that may exist across a glacier surface (e.g. Smith *et al.*, 2016). The turbulent footprint referred to above is the source region for the turbulent fluxes received at a given location. It represents the upwind area that influences and contributes to the observed fluxes, and hence, the surface properties that modulate turbulence generation. Broadly speaking, the turbulent footprint for fluxes measured at a given height will extend upwind by a distance of roughly 100 times the measurement height (Burba, 2013).





Efforts have been made in previous boundary-layer studies over different land surfaces to determine momentum roughness length values for large areas, including over forestry, scrubland, and outwash plains (e.g. Nield *et al.*, 2013; Li *et al.*, 2017). A range of remote sensing techniques have been implemented in such studies, including the use of light detection and ranging (LiDAR) systems. Paul-Limoges *et al.* (2013) used digital elevation models (DEMs), obtained from airborne

LiDAR, to estimate $z_{0v}$ values over a harvested forest surface ($z_{0v}$ = 0.13 m), and found good agreement with corresponding EC-derived values ($z_{0v}$ = 0.12 m). Similar studies on mountain glaciers are extremely rare. Smith *et al.* (2016) used terrestrial-based structure-from-motion photogrammetry and laser surveying to generate a distributed map of $z_{0v}$ estimates for a glacier. Meteorological-based evaluation of the returned $z_{0v}$ estimates was not carried out, however.

The scalar roughness lengths ($z_{0t}$ and $z_{0q}$) are commonly estimated in SEB studies using a fixed ratio with $z_{0v}$, and are

generally assumed to be equal to or one to two orders of magnitude smaller than the momentum roughness length (e.g. Hock and Holmgren, 2005; Sicart *et al.*, 2005; Hoffman *et al.*, 2008). Molecular diffusion controls the rate of scalar transfer with a surface, and having a smaller spatial scale than the form drag processes driving momentum transfer, it is likely that the scalar roughness lengths would be smaller (Beljaars and Holtslag, 1991). The persistence of this ratio with time is uncertain, however. Surface renewal methods have been implemented in some studies (e.g. Andreas, 1987; Smeets and van den

Broeke, 2008), where variation in this ratio is described as a function of the roughness Reynolds number $R_*$. Changes in mean air temperature and relative humidity have also been proposed as drivers of scalar roughness length variation (e.g. Calanca, 2001; Park *et al.*, 2010).

The initial goal of this study is to obtain *in situ* values of the momentum and scalar roughness lengths from multiple locations over several seasons. EC-observed data will be implemented into the bulk aerodynamic method to derive these

values. The temporal variability of roughness lengths on a glacier will be examined, and the transferability of values between location and years will be assessed. Commonly assumed values and ratios from the literature will be compared with the obtained data, and predictive relationships for the scalar roughness lengths will be tested. The second goal of this study is to develop remote methods for estimating momentum roughness lengths for a glacier surface, which would facilitate SEB modelling for glaciers without *in situ* observations, and distributed modelling for glaciers with point measurements only.

Digital elevation models will be obtained for each study location and will be used to provide surface height data for the two roughness methods developed in the study. Turbulence footprint modelling will be employed in one of these methods to identify the region of the glacier surface influencing the EC-derived roughness length values. The estimates from both remote methods will then be compared with those from corresponding *in situ* observations.



## 2 Data and Methods

### 2.1 Field Campaign

Observations were carried out over three melt seasons (2014–2016) on two glaciers in the Purcell Mountains of British Columbia, Canada (Fig. 1). Nordic Glacier (51°26' N, 117°42'W) is a small (~5 km$^2$), north-facing glacier, ranging in

elevation from 2,000 m to 2,900 m above sea level (a.s.l.), approximately. An automatic weather station (AWS) was installed in the ablation zone of the glacier through July and August 2014 (NG14). Conrad Glacier (50°49' N, 116°55'W) is located 87 km to the southeast of Nordic, with an area of ~15 km$^2$, and an elevation range of 1,800 m to 3,200 m a.s.l., approximately. A total of four AWS deployments were executed on Conrad during 2015 and 2016; two stations in the ablation zone from July to September 2015 (CG15-1 and CG15-2), and one in both the ablation (CG16-1) and accumulation

(CG16-2) zones from June to August 2016 (Table 1). An exposed ice surface was present during observations at NG14, CG15-1, CG15-2, and for most of the observation period at CG16-1, while a snow surface was present throughout at CG16-2, and for the first 10 days at CG16-1. A transitional snow surface was present for the first four days at NG14, with partial snow cover diminishing to a fully bare ice surface.

### 2.2 AWS

The AWS developed for this project (see Fitzpatrick *et al*., 2017) was equipped with an array of meteorological and glaciological sensors to observe the complete SEB, with additional sensors added to the stations each year (Table 2). Open and closed path eddy covariance (OPEC and CPEC) systems were used in this project to observe the turbulent heat fluxes, with both forms installed on the same station, in some cases (CG15-1 and CG16-1). Both systems were comprised of a 3D sonic anemometer, and an infrared gas analyser; the OPEC analyser has a sample space that is open to passive air flow,

while the CPEC analyser has a closed sample space into which air is drawn using a pump. Implementing these methods together helped minimise gaps in the turbulence dataset (OPEC analysers are susceptible to errors during precipitation), and enabled a comparison of their values and performance in a glacial environment. The EC data was recorded in raw 20Hz format, with observations from the remaining sensors stored in one-minute averages.

The meteorological sensors were housed on a four-legged quadpod, which provided a stable platform (verified by an

inclinometer sensor) that lowered as the ice melted, and maintained a constant height of the sensors above the surface. EC measurements were carried out at a constant height (~2 m at each station) to avoid substantially varying the turbulence footprint area and to reduce the risk of elevating the sensor above the turbulence coupled with the surface (Burba, 2013; Aubinet, 2008). The installation site for each station was selected based on the criteria of a relatively uniform upwind footprint and slope angle, so as to minimise the corrections required in the EC (and radiation) data processing. The EC

systems were installed on the upslope side of each station, so as to be the first point of contact with the prevailing wind (downslope), and to help minimise flow distortion. Time lapse cameras at each location were used to observe the surface and atmospheric conditions over a season, and to monitor station behaviour. Over the three melt seasons, the stations performed



well, operating continuously over each study period. The solar power systems for the stations had been designed to have sufficient battery storage for approximately a week of operation without sufficient recharge (due to persistent overcast conditions or covering of the solar panels by snow/ice.). If battery voltages dropped below a critical level, the system was designed to restrict power supply to the higher consuming sensors (e.g. CPEC system) to ensure continued operation of the

bulk of the instruments, and to allow the batteries to recharge. This occurred at only one station, CG16-2 in the accumulation zone, after consecutive periods of snowfall and persistent low cloud, resulting in four intermittent gaps in the CPEC dataset (28% of total observation time).

## 2.3 LiDAR

Airborne LiDAR was employed to obtain high resolution topographic data over each of the study locations, using a Riegl

580 laser scanner and dedicated Applanix PosAV 910 Inertial Measurement Unit. In general, flights were performed over Nordic and Conrad glaciers twice per year (Table 3), close to the end of the winter and summer seasons (April and September), as part of an ongoing mass balance survey of the study glaciers (B. Pelto, unpublished data). By analysing the altimetry data from these times of the year, it was hoped that the variation in surface roughness due to the transition from a snow-covered to bare ice surface could be captured. In addition, the repeat mapping of each location from one year to the

next would help identify the persistence in surface roughness. In 2014, April flights were not performed over the glaciers (a July flight was performed over Nordic), while in 2015, the September flight over Conrad captured usable data for the accumulation zone, only.

## 2.4 Data Treatment

### 2.4.1 Eddy Covariance Data

Prior to calculating observed values for the turbulent heat fluxes and roughness lengths, the raw (20 Hz) EC data were passed through a series of preprocessing steps using the EddyPro data package (LI-COR, 2016). These steps are described in detail in Fitzpatrick *et al*. (2017), but a summary of the main techniques is provided below. A planar fit coordinate rotation method (Wilczak et al., 2001) was applied to all of the sonic anemometer data to account for misalignment of the $z$ axis of the sensor with the $w$ component of the mean air flow. For the OPEC water vapour measurements, the Webb-Pearman-

Leuning correction (Webb *et al*., 1980) was used to correct for the density effects of air temperature fluctuations, while readings from periods affected by precipitation on the analyser windows were removed. These corrections were not required for the CPEC water vapour data. The turbulence data were averaged over 30-minute blocks, and the calculated fluxes were filtered using quality tests for steady state and developed turbulent conditions, following Mauder and Foken (2004).



### 2.4.2 LiDAR Data

The trajectories of each LiDAR flight had been previously post processed using a network of permanent GPS base stations in British Columbia. The positional uncertainties of the flight trajectories were typically better than 5 cm, with the total uncertainty in the processed LiDAR point clouds better than ±10 cm, while the average point density for the LiDAR surveys over the ice-covered terrain was 1–2 laser shots per m² (B. Pelto, unpublished data). LAStools (Isenburg, 2006) was utilised to classify the LiDAR data into ground and non-ground laser returns. The ground returns were subsequently gridded into DEMs with a 1 m² grid cell, the grid lines aligned with true north and east.

### 2.5. In Situ Roughness Length Values

Roughness length values were calculated by implementing EC data into the bulk method, with separate values calculated for OPEC and CPEC systems when both sensors were used at the same station:

$$z_{0v\_ec} = exp\left[-\kappa\frac{u_{ec}}{u_{*ec}} - \psi_m\left(\frac{z}{L_{ec}}\right)\right]z\,, \tag{1}$$

$$z_{0t\_ec} = exp\left[-\kappa\frac{T_{ec}-T_s}{\theta_{*ec}} - \psi_h\left(\frac{z}{L_{ec}}\right)\right]z\,, \tag{2}$$

$$z_{0q\_ec} = exp\left[-\kappa\frac{q_{ec}-q_s}{q_{*ec}} - \psi_h\left(\frac{z}{L_{ec}}\right)\right]z\,, \tag{3}$$

where $\kappa$ is the von Kármán constant (0.4), $z$ is the sensor height, and $u_{ec}$, $T_{ec}$, $q_{ec}$, $u_{*ec}$, $\theta_{*ec}$, and $q_{*ec}$ are the 30 minute EC-observed values for mean wind speed, air temperature, specific humidity, friction velocity, and the surface layer scales for temperature and specific humidity, respectively (Conway and Cullen, 2013). $\psi_m\left(\frac{z}{L_{ec}}\right)$ and $\psi_h\left(\frac{z}{L_{ec}}\right)$ are the vertically integrated stability functions for momentum and heat (Beljaars and Holtslag, 1991; Dyer, 1974), where $L_{ec}$ is the Monin-Obukhov length. Glacier surface specific humidity $q_s$ is calculated from atmospheric pressure $p$, and the surface vapour pressure ($e_s$) which is assumed to be at saturation at the glacier surface temperature $T_s$ ($q_s = 0.622\,e_s/p$). To minimise potential errors and to obtain roughness lengths representative of the conditions at each site, an extensive series of filters were applied to the 30-minute values (see Fitzpatrick *et al.*, 2017, for full details). These filters included a 90° wind direction window centred on the main axis of the EC sensor (to minimise the influence of flow distortion due to the station structure), minimum values for wind speed (> 3 m s⁻¹) and $u_{*ec}$ (> 0.1 m s⁻¹), minimum differences between measurement and surface height values of air temperature (> 1°C) and vapour pressure (> 66 Pa) (Calanca, 2001; Conway and Cullen, 2013), a minimum scalar roughness length value of 1 x 10⁻⁷ m based on the mean free path length of molecules (Li *et al.*, 2016), a precipitation filter, and a test for stationarity of the turbulence (Foken, 2008). Only roughness length values calculated during near-neutral stability conditions ($-0.1 < \frac{z}{L_{ec}} < 0.2$) were retained, to minimise the uncertainty associated with the stability functions applied in Eq. (1–3) during non-neutral conditions (Smeets and van den Broeke, 2008; Conway and Cullen, 2013).



### 2.5.1 Scalar Roughness Length Modelling

The scalar roughness lengths from Eq. (2) and (3) were compared with values from the surface renewal models of Andreas (1987) and Smeets and van den Broeke (2008), where the ratio of the scalar ($z_{0s}$) and momentum roughness lengths are expressed as a function of the roughness Reynolds number $R_*$:

$$R_* = \frac{u_* z_{0v}}{\nu}, \tag{4}$$

$$ln\left(\frac{z_{0s}}{z_{0v}}\right) = b_0 + b_1 ln(R_*) + b_2 ln(R_*)^2 . \tag{5}$$

$\nu$ is the kinematic viscosity of air (1.5 x $10^{-5}$ $m^2$ $s^{-1}$), and the EC-derived roughness lengths (Eq. 1–3) were used to populate $z_{0v}$ and $z_{0s}$. The values of the empirical coefficients ($b_0$, $b_1$, and $b_2$) change for smooth ($R_* \leq 0.135$), transitional ($0.135 < R_* < 2.5$), and rough ($R_* \geq 2.5$) flow regimes, and between models.

### 2.6 Remote Momentum Roughness Length Estimation

The set of 1 x 1 m grid cell DEMs obtained for the study glaciers from the LiDAR data were utilised to remotely estimate momentum roughness length values. Estimates were determined at the location of each station using the DEMs from the same year the station was in place, and compared with the EC-derived $z_{0v\_ec}$ values. September DEMs were used to estimate roughness length values for bare ice surfaces, and April DEMs for snow-covered surfaces (both the April and September DEMs at CG16-2 in the accumulation zone represent a snow-covered surface). The DEM for Nordic Glacier in July 2014 was used to estimate roughness lengths for the transitional snow-ice surface at NG14. The estimation of $z_{0v}$ was also repeated on DEMs from periods without a station present at that location to allow for an examination of the temporal variation of roughness properties at each site over the three years. Two methods were developed in this study, referred to as the (i) block and (ii) profile methods. Both methods assume that a DEM with a 1 x 1 m grid cell can adequately resolve the scale of the surface features that have the primary influence on roughness length. Where airflow encounters a dense distribution of roughness elements (as can be present on an ablating glacier surface), the flow is likely to experience wake-interference or skimming (Wieringa, 1993), reducing the relative influence of smaller scale roughness features on $z_{0v}$ (Smeets *et al.*, 1999), and increasing the influence of elements that are potentially resolvable at the DEM scale.

Both methods draw on the empirical theory of Lettau (1969) for the estimation of $z_{0v}$ from microtopography measurements:

$$z_{0v} = 0.5 h^* \frac{s}{S}, \tag{6}$$

where $h^*$ is the average effective height of the roughness elements above the surface, $s$ is the average crosswind silhouette or face area of the roughness elements encountered by oncoming air flow, $S$ is the lot area, equal to the total area of the site divided by the number of roughness elements on its surface, and the value 0.5 represents an average drag coefficient. The original application of the above theory assumes that the surface is composed of regularly spaced roughness elements of similar size and shape, an assumption that may not always hold for a glacier surface.



### 2.6.1 Block Estimation

The first method developed in this study to estimate $z_{0v}$ aimed to account for the variation in shape and distribution of roughness elements on a glacier surface. First, the form drag generated by the features on an individual portion or block of the surface was estimated, before combining the influence of each portion over a footprint to determine the momentum

roughness length value for a given downwind location. Similar methods were proposed and evaluated by Kondo and Yamazawa (1986) for estimating $z_{0v}$ over irregular surfaces. To account for the often dense distribution of roughness elements on a melting glacier surface, and the effects of this distribution on airflow, the block method developed here also considers the relative height differences and potential sheltering influence of neighbouring features on the surface.

As the method would be evaluated using roughness measurements derived from the EC systems, it was applied to subareas

of each DEM that contained the potential turbulent footprint for a given station. Each subarea was 2,000 x 2,000 m in dimension, and centred on the grid cell containing the station site. For each grid cell in the subarea, a one-cell-thick border was selected around the cell of interest, creating a 3 x 3 m block of cells (Fig. 2), representing a roughness element and its surrounding area of influence. A localised drag value ($F_{D\_local}$) was estimated for each block, by utilising Eq. (6), and building on the methods of Smith *et al*. (2016). The heights of the cells in the block were detrended for the mean slope of the

glacier in the region of the station, as it was assumed that mean airflow was parallel to this plane. The height values within the block were normalised, and the mean height of all the cells above the zero plane was assigned to $h_b{}^*$. A value for $s_b$ was calculated for each cardinal wind direction, as follows. The heights of the first line of cells in the block perpendicular to the oncoming wind ($h_{i1}$) set the base levels for the silhouette area, and the maximum height of the cells in each row set the upper level. The sum of the silhouette areas of each row was then assigned to the $s_b$ value for that block and wind direction:

$$s_b = \sum_{i=1}^{n} \max(h_{ij}) - h_{i1},\tag{7}$$

where $n$ is the number of rows. The area of the block was assigned to the value for $S_b$. $F_{D\_local}$ values were then calculated for each of the four cardinal wind directions for each grid cell; the block in Fig. 2 shifting by one cell each step:

$$F_{D\_local} = 0.5 h_b{}^* \frac{s_b}{S_b}.\tag{8}$$

A range of border thicknesses around each grid cell, from one to five cells (3 x 3 to 11 x 11 m block area), was also

implemented to test the performance sensitivity to this choice.

To estimate a momentum roughness length value at the location of a station, the effective influence of the $F_{D\_local}$ values over the entire footprint must be determined. The flux footprint of the turbulence observed at each station was estimated using the model of Kljun *et al*. (2015). This model involves a two-dimensional parameterisation of a more complex, backward Lagrangian particle dispersion model (the LPDM-B model in Kljun *et al*., 2002). In the above study, the

parameterisation was developed and evaluated for a wide range of boundary layer conditions and surface types, and was shown to agree with the footprint estimates of the more complex model. To estimate the footprints for the glacier stations in this study, EC-observed values for mean wind speed and direction, $z_{0v\_ec}, L_{ec}, u_{*ec}$, and the standard deviation of lateral wind velocity were implemented into the parameterisation. Flux footprint maps were generated from the model, with a 1 x 1





m grid cell and total area of 2,000 x 2,000 m, centred on the station location, to match the selected DEM subareas. Each grid cell was assigned a flux footprint value ($f_c$), representing its normalised contribution to the turbulent flux observed at the station. Maps were generated for every 30-minute period in the EC data, from which an average seasonal footprint for the station was determined. For stations with two EC systems, separate footprint maps were generated for each to investigate

sensitivity to the observation method.

The seasonal flux footprint map for a given station (or EC system) was overlaid over the corresponding $F_{D\_local}$ values for the wind direction of interest. The $F_{D\_local}$ value for each grid cell was then weighted by its flux footprint contribution, and summed over the subarea to obtain $z_{0v\_bloc}$:

$$z_{0v\_bloc} = \sum_{i=1}^{n} F_{D\_local_i} f_{c_i} , \qquad (9)$$

where $n$ is the number of grid cells in the subarea. This process was then repeated for the DEMs available from each season. Standard error propagation methods were used to calculate the uncertainty in $z_{0v\_bloc}$ by considering the uncertainties in the LiDAR height data (< ±0.1 m) and the normalised mean square error in the $f_c$ values from the footprint model (0.48; Kljun *et al.*, 2015).

The primary application of a remote technique to estimate momentum roughness lengths would be to obtain values for where

*in situ* observations are not available, and therefore, where the turbulent flux footprint for a given site is unknown. $z_{0v\_bloc}$ values were first calculated with EC-derived footprints, as above, to evaluate the effectiveness of the local form drag estimation (Eq. 8). To test the performance of the block method in situations when EC data is not available, the observed turbulent footprints were then replaced with a series of assumed footprint areas at each site and applied to the corresponding $F_{D\_local}$ values to calculate $z_{0v\_bloc}$.

**2.6.2 Profile Estimation**

The second method developed in this study takes a profile-based approach to estimating momentum roughness lengths, and aims to identify the length scales relevant to form drag over that profile, rather than using the element by element approach of the previous technique. As with the block method, the first step was to detrend the surface height values for the mean slope of the glacier. Beginning with roughness estimation for the downslope (southerly) wind direction, a profile of grid cells

was selected from a given DEM along the glacier slope; 600 m in length, one grid cell wide, and centred on the location of a station. A linear trend was fitted to this profile to identify the slope, and the trend was then removed from the original height data (Fig. 3a-b). This step was repeated for 50 parallel profiles on either side of the central 'station' profile (101 profiles, in total). The next step was to determine the scale of the features relevant to form drag, that is, the features that act as obstacles to air flow, and to remove large scale surface features or waves which air flow may follow rather than be impeded by. The

power spectrum was calculated for the detrended profile, and a cut-off wavelength was visually identified between large and small scale features. In Fig. 3c, an example of the mean power spectrum over 101 detrended profiles is shown in log-log for CG16-1 in September 2016. In this case, a cut-off wavelength of $\lambda_0 = 35$ m was visually identified from the plot as





differentiating between wavelengths with large or small power or amplitude. With the cut-off wavelength identified, a fast Fourier transform (FFT) high pass filter was applied to the detrended profile to remove the large wavelengths (Fig. 3b), and to obtain a filtered profile. The filtering was performed in the wavenumber ($k$) domain with the following steps: (i) FFT was applied to the detrended profile $h(y)$ in Fig. 3b to get $H(k)$; (ii) $H(k)$ was modified by setting its values to zero for $k < 2\pi/\lambda_0$;

(iii) an inverse FFT was applied to the modified $H(k)$ to get the filtered profile $h(y)$ in Fig. 3d. Finally, Lettau's method (Eq. 6) was applied to the filtered profile to estimate a value for momentum roughness length. $S$ was calculated as the width of the profile ($w = 1$ m) multiplied by the length of the fetch ($L_F$) upwind of the station. A range of values for $L_F$ were applied from $\lambda_0$ to $2\lambda_0$ in 1 m increments. The height of the grid cells along a given fetch was assigned to an array from $h_0$ to $h_N$, where $N$ is the number of grid cells in the fetch, and the standard deviation of the height array along $L_F$ was assigned to $h^*$. A value

for $s$ was obtained from the sum of the height differences between adjoining grid cells:

$$s = w \sum_{j=1}^{N} |h_j - h_{j-1}|, \tag{10}$$

and substituted into Eq. (6), with $S$ and $h^*$, to estimate a momentum roughness length value for a given fetch ($z_{0v\_fetch}$). The mean of the $z_{0v\_fetch}$ values from $L_F = \lambda_0$ to $2\lambda_0$ was then assigned to the momentum roughness length for the station grid cell ($z_{0v\_prof}$).

To examine roughness length variability in the vicinity of the station grid cell, and to determine the uncertainty in the presented results, the above process was repeated for all grid cells in the 101 x 101 m area upwind of the station (i.e. 50 m either side of the station profile). The profile method was also applied over a range of angles in addition to the prevailing downslope, southerly direction, to examine the effects of changing wind direction on momentum roughness length (Fig. 4). To do so, the x-y grid matrix of a patch of grid cells (101 m wide and 351 m long, containing the station site) was multiplied

by a rotation matrix (in 5° increments between 90° and 270°). The height values from the DEM grid cells were then bi-linearly interpolated to the rotated grid to derive new rotated height values. A value for $z_{0v\_prof}$ was then calculated as above for profiles in line with the long axis of the patch, for each 5° increment in direction.

The sensitivity of the profile method to the use of a DEM with a finer (1 x 0.1 m) or coarser resolution (3 x 3 m) than the original 1 x 1 m DEM was tested. As a 1 x 0.1 m DEM could not be derived from the LiDAR data, a synthetic test surface

was created using data from microtopography profile measurements obtained at CG16-1 at the end of the melt season. Four surface height profiles, 2 m in length and with 0.1 m resolution, were obtained at distances of 10 m, 50 m, 100 m, and 150 m upwind of the station (Fig. 5a). The profiles were taken perpendicular to the prevailing wind direction (downslope), and measured using a 2 m snow probe, horizontally laid on the surface and allowed to partially melt in place. The long axis of the probe was set as the zero plane, and the height of the surface was measured relative to this level at 0.1 m spacings.

Height variability parallel to the downslope direction was expected to be smaller than in the parallel direction which crosscuts supraglacial channels on the surface. Therefore, in the absence of microtopography measurements in this direction, the profile from the cross-slope direction with the smallest variance i.e. the 10 m upwind profile (Fig. 5b), was used to represent the slope-parallel variance. This 2 m profile was demeaned at a 1 m interval and lined up in a repeated sequence to



obtain an extended (600 m long) synthetic microtopography profile. The final test profile was constructed by adding this extended synthetic profile to the detrended profile in the downslope wind direction from the 1 x 1 m DEM. The same synthetic profile was added to the detrended profiles from each side of the station, at 1 m distance apart, yielding the synthetic 1 x 0.1 m DEM. The 3 x 3 m DEM was created by applying a 2-D smoothing of the original 1 x 1 m DEM, using a

3-point running mean in both $x$ (Easting) and $y$ (Northing) directions. The profile method was then applied to both the 1 x 0.1 m and 3 x 3 m DEMs for the 600 x 101 m area upwind (slope-parallel) of the station, using the same steps as outlined previously. The same threshold wavelength, $\lambda_0$ = 35 m, was used to filter the profiles. Figure 5c displays examples of filtered profiles, $h(y)$, as derived from the three DEM resolutions.

## 3 Results

### 3.1 EC-Derived Roughness Lengths

The geometric means of the roughness length values calculated from each EC dataset are presented in Table 4, with separate $z_{0v\_ec}$ values for periods with snow and ice surfaces. Each of the observed 30-minute roughness length datasets were found not to have a normal distribution (using one-sample Kolmogorov-Smirnov tests), but one that was approximately log-normal. For presenting mean EC-derived values in the remainder of this study, geometric means are used to avoid

excessively weighting the larger roughness values (Andreas *et al.*, 2010). Stable atmospheric conditions persisted over the glaciers for much of each season, limiting the number of suitable 30-minute periods for roughness calculation after application of the filters discussed in Sect. 2.5 (number of available measurements presented in Table 4). Across all test sites, $z_{0v\_ec}$ had a mean of 2.3 mm and 1 mm for ice and snow, respectively, while the scalar roughness lengths had mean values of 0.05 mm for $z_{0t\_ec}$ and 0.11 mm for $z_{0q\_ec}$. Where OPEC and CPEC systems were used on the same station, the OPEC

system returned slightly larger mean $z_{0v\_ec}$ values (2.8 mm and 1.4 mm, respectively). Mann-Whitney U tests applied to the 30-minute roughness values from CG15-1 rejected the null hypothesis that the $z_{0v\_ec}$ values from the OPEC and CPEC systems had the same distribution (p < 0.01), but the hypothesis could not be rejected for the scalar values (p > 0.5).

The ice $z_{0v\_ec}$ values were within the expected range for moderately rough glacier ice (Brock *et al.*, 2006). Where measurements were repeated in the same area a year apart (CPEC observations on CG15-2 and CG16-1), persistence in the

mean ice roughness length values was noted (0.86±7.4 mm, and 0.74±6.4 mm, respectively), with a failure to reject the hypothesis of equal distributions (p = 0.16). Within a season, substantial variability was noted in the 30-minute $z_{0v\_ec}$ values for each ice surface (Fig. 6a), but with no evident trend in $z_{0v\_ec}$ due to changes in surface roughness over time. Mean momentum roughness lengths for snow were also within previously observed values on glacier surfaces, with a particularly large mean value observed at CG16-2 in the accumulation zone (2.4±16 mm). Extensive variability was also present in the

30-minute $z_{0v\_ec}$ values for CG16-2 (Fig. 6b), with a general increasing trend in roughness over the season. Across all stations and seasons, substantial variability was noted in the mean scalar roughness lengths, with $z_{0q\_ec}$, in particular,



showing a range of two and a half orders of magnitude. $z_{0t\_ec}$ exhibited less variability (~one order of magnitude), with similar mean values observed for CG15-2 and CG16-1 (0.03±0.28 mm and 0.05±0.29 mm), and a failure to reject the null hypothesis of equal distributions (p = 0.11).

The ratios of the 30-minute EC-determined scalar roughness lengths to $z_{0v\_ec}$ were expressed as a function of $R_*$ using the
data from all stations and seasons (Fig. 7). These values were compared with the surface renewal models of Andreas (1987) and Smeets and van den Broeke (2008). The seasonal mean ratios and $R_*$ were also compared with these models. In general, the roughness ratios were shown to decrease with increasing $R_*$, with substantial scatter in the 30-minute values. The seasonal mean $\frac{z_{0t}}{z_{0v}}$ ratios were in line with the output of the Andreas (1987) model (r 0.81; p <0.05), with greater scatter in the $\frac{z_{0q}}{z_{0v}}$ values (r = 0.2), while both sets of ratios were underestimated by the Smeets and van den Broeke (2008) model.

**3.2 Momentum Roughness Length from LiDAR**

**3.2.1 Block Method**

$F_{D\_local}$ maps were generated from LiDAR-derived DEMs using the block estimation method (Fig. 8a-b) for all available years and seasons, and for each of the four cardinal wind directions. Substantial variation in $F_{D\_local}$ was observed across each glacier surface, ranging from $10^{-4}$ m for snow-covered grid cells to $10^{-0.5}$ m for large crevasses. Figure 9 displays the
seasonal turbulent flux footprint maps generated using the model of Kljun *et al*. (2015) for each EC sensor deployment. In general, the fluxes were sourced from regions to the south of each station, in line with the prevailing downslope winds at each site. Over 80% of flux contribution came from an area within 200 m upwind of each station, with concentrated peak source regions 15–20 m upwind, on average. The flux footprints of each EC dataset were merged with the corresponding $F_{D\_local}$ maps (Fig. 8c), producing a series of $z_{0v\_bloc}$ values for each site. As stated, wind direction was predominately from
the south during each station deployment, so the roughness estimates for this wind direction (Table 5) are used for comparison with the EC-derived values. The influence of wind direction on the roughness length estimates is discussed in Sect. 3.3 and 4.1.3.

The mean uncertainty in the $z_{0v\_bloc}$ values, estimated from propagation of the errors in the LiDAR and flux footprint values, was ±0.53 mm. Where OPEC and CPEC systems were used simultaneously on the same station (CG15-1 and CG16-1),
virtually identical $z_{0v\_bloc}$ values were returned when their flux footprints were applied. Therefore, only one set of values is presented for each station in Table 5. Mean $z_{0v\_bloc}$ values for ice and snow surfaces, over all sites and seasons, were 3.1 mm and 0.6 mm, respectively, with strong persistence in site roughness values from one year to the next. A range of assumed footprint areas were also applied to the $F_{D\_local}$ maps to determine the effectiveness of the method in the absence of observed footprint data. Applying equal weighting to $F_{D\_local}$ values in a 101 x 101 m area directly upwind of a site ($f_{c\_100}$) was found
to return roughness values close to the $z_{0v\_bloc}$ and $z_{0v\_ec}$ values, in most cases (Table 5).



As previously stated, the sensitivity of roughness length estimation to the selected block size was tested by varying the border thickness around the grid cell of interest. Overall, increasing the block area was found to lead to an increase in estimated roughness length for a given footprint, with a border thickness of 1 cell (3 x 3 m block area) returning roughness lengths closest to the EC-derived values at all stations (e.g. CG16-1 ice $z_{0v\_bloc}$ = 1.6, 1.9, 2.0, 2.2, and 2.4 mm for an

increasing border thickness range of 1-5 cells).

### 3.2.2 Profile Method

The detrending and filtering of the surface height data, as shown in Fig. 3, was performed for downslope profiles at each station site using the DEMs for all available years and seasons. The same approximate value for the cut-off wavelength ($\lambda_0 \approx$ 35 m) was identified at each station site. $z_{0v\_prof}$ values were then estimated for each station location, and for each grid cell

in a 101 x 101 m upwind area (Fig. 10a), from all corresponding DEMs. Table 5 presents the $z_{0v\_prof}$ values for each station and LiDAR flight. Mean $z_{0v\_prof}$ values for ice and snow surfaces, over all sites and seasons, were 4.3 mm and 1.1 mm, respectively. Where repeated over the same location, the $z_{0v\_prof}$ values displayed substantial differences from one year to the next over ice surfaces (up to 5 mm), in contrast to the noted $z_{0v\_bloc}$ persistence.

Fig. 10b displays the $z_{0v\_prof}$ values derived for the downslope profiles from the original DEM (1 x 1 m), and from the

higher (1 x 0.1 m) and lower (3 x 3 m) resolution DEMs constructed for sensitivity testing. Roughness values are presented for the station location at CG16-1 and for the grid cells 50 m to the east and west of the station. The same pattern of spatial variability in $z_{0v\_prof}$ across the grid cells was captured with each DEM, but with substantial differences in magnitude. On average, the 3 x 3 m DEM yielded $z_{0v\_prof}$ values one order of magnitude smaller than the original 1 x 1 m DEM. This result is expected since the original surface has been smoothed, and the relevant scales of the roughness elements may not be

adequately resolved in the 3 x 3 m DEM. Applied to the 1 x 0.1 m DEM, the profile method yielded roughness values, on average, a half order of magnitude larger than those for the 1 x 1 m DEM. The primary reason for differences in $z_{0v\_prof}$ values with changing DEM resolution was the difference in $s$ values (Eq. 10). While $h^*$ values remained almost unaltered for different resolutions, the $s$ values changed by > 50%, resulting in large changes in $z_{0v\_prof}$.

The first-order estimate of surface variability from the microtopography survey may overestimate the variability in the

downslope wind direction in the 1 x 0.1 m DEM. To test for this, the amplitude of the synthetic microtopography profiles was reduced by a factor 10 (from dm to cm scale) and $z_{0v\_prof}$ recalculated. The resulting roughness length values were reduced and matched more closely the original $z_{0v\_prof}$ from the 1 x 1 m DEM, however, still yielding up to 10% larger values than original (Fig. 10b).

### 3.2.3 In Situ vs Remote Methods

The estimates from both DEM-based roughness methods were compared with the EC-derived values (Fig. 11 and Table 6). In cases where LiDAR data were not available from the same year a station was in place, the averages of the roughness





estimates from the two other years were utilised for the comparison. Overall, estimates from both DEM-based roughness methods provided values for ice and snow surfaces in line with previous observations on glacier surfaces (Brock *et al.*, 2010), and were generally within 2 mm and 0.2mm (< half order of magnitude) of the corresponding $z_{0v\_ec}$ observations over ice and snow, respectively. Over ice surfaces, the $z_{0v\_bloc}$ values were slightly smaller than the corresponding $z_{0v\_prof}$

values (mean values of 3.1 mm and 4.3 mm, respectively), and tended to align more closely with the $z_{0v\_ec}$ estimates (mean of 2.3 mm). For the snow surface at CG16-2 in the accumulation zone, the mean roughness lengths from both DEM methods (0.4 mm) substantially underestimated the $z_{0v\_ec}$ value (2.4 mm). Potential causes for this deviation will be discussed in Sect. 4.1.2. For the transitional snow/ice surface present at NG14 during the first four days of observations, the $z_{0v\_bloc}$ and $z_{0v\_prof}$ values from the July 2014 flight (4.5 mm and 6.8 mm) aligned more closely with the mean $z_{0v\_ec}$ value for ice over

the season (4.5±28.8 mm) than with the $z_{0v\_ec}$ value obtained during the four day period (0.5±3.0 mm). The mean $z_{0v\_ec}$ value for this period, however, was based on a very limited number of EC observations after filtering (n=16) with substantial scatter.

**3.3 Wind Direction and Momentum Roughness Length**

The 30-minute EC data and the rotated $z_{0v\_prof}$ values were used to examine the influence of wind direction on the effective

roughness length at each location. It should be restated at this point that the $z_{0v\_ec}$ values had been filtered to remove values when wind direction was beyond ±45° of the main axis of the EC sensor to minimise the influence of flow distortion due to the station structure. Therefore, only a limited direction window is available in the $z_{0v\_ec}$ data over which to examine this dependence. For the ice surface of CG16-1, $z_{0v\_ec}$ values were observed to increase and become more scattered as the wind direction veered towards the southwest, a pattern that was also detected in the $z_{0v\_prof}$ (Fig. 12a). Similar behaviour was

noted at the same location in 2015 (CG15-2), with greater variation in $z_{0v\_ec}$ with wind direction (Fig. 12b).

The rotated $z_{0v\_prof}$ values were also used to examine a wider angle of wind direction than was possible with the $z_{0v\_ec}$ data. Fig. 13 displays the $z_{0v\_prof}$ values in 5° increments in wind direction between 180° and 270° for an April (snow) and September (ice) surface at each station. The magnitude of roughness length variation with direction was greatest over ice surfaces. For the three stations in Conrad's ablation zone, $z_{0v\_prof}$ was observed to increase as wind direction approached a

cross-glacier orientation (east or west), while at NG14, a pronounced increase in roughness was noted over the ice surface at 240°. The snow surfaces at CG16-2 in April and September presented very similar roughness profiles with wind direction, with slightly larger $z_{0v\_prof}$ in the autumn. The apparent peaking in $z_{0v\_prof}$ over CG16-2 at 90°, 180°, and 270° is likely the result of an artificial reduction in roughness at all other angles due to the smoothing of the DEM when the height values were bi-linearly interpolated to the rotated grid. The roughness values at 90°, 180°, and 270° are calculated from the original

DEM, without the need for interpolation, and the effect of this appears to be most visible in the smaller magnitude $z_{0v\_prof}$ values over the snow surface.





## 4 Discussion

### 4.1 Spatial and Temporal Variance of $z_{0v}$

#### 4.1.1 Ice Surfaces

Variation in both the $z_{0v\_ec}$ and DEM-based roughness length values was noted across test sites with a melting glacier ice
surface (e.g. 4.5 mm and 0.7 mm for mean $z_{0v\_ec}$ at NG14 and CG16-1, respectively). An assumed $z_{0v}$ value for ice (e.g.
1mm), applied uniformly to all locations in this study, would have substantially misrepresented the surface roughness
characteristics, and the resulting turbulent flux parameterisations. In the case of NG14, implementing the commonly
assumed $z_{0v}$ value for ice of 1 mm in the bulk parameterisation of turbulent heat fluxes, rather than the mean observed value
of 4.5 mm, would result in a ~20% reduction in the mean estimated fluxes. Furthermore, stations throughout the study were
installed in secure regions of the glaciers with relatively smooth and uniform surfaces, and away from crevasse fields and
glacier margins where the surface drag on airflow would be higher (Fig. 8). Therefore, the true range of roughness length
values over the entire surface of the study glaciers would be greater than that represented by the values estimated for the
station locations. Smith *et al*. (2016) detected a $z_{0v}$ range of over three orders of magnitude across a small (~1 km$^2$)
mountain glacier (Kårsaglaciären in Sweden).

Over the study period, the mean momentum roughness length estimates for ice at each site showed little temporal variance
from one year to the next. This persistence in seasonal ice roughness values may allow for the use of $z_{0v}$ estimates from pre-
existing EC or DEM campaigns at a site of interest. The period of validity of these estimates may vary, however, depending
on the surfaces processes of each glacier. Within a single melt season, there was substantial scatter observed in the 30-
minute $z_{0v\_ec}$ values (Fig. 6a). Changes in momentum roughness length due to the evolution of the ice surface through the
season were not evident in the $z_{0v\_ec}$ values, however. Previous glacier roughness studies (e.g. Sicart *et al*., 2014) have also
noted persistence in $z_{0v}$ despite extensive ice melt. Smith *et al*. (2016) noted that this persistence was most evident over ice
surfaces with defined melt features, such as supraglacial channels, similar to the ice surfaces of this study. While estimated
using EC-observed data, the $z_{0v\_ec}$ calculations are still derived from the bulk aerodynamic method (Eq. 1). Extensive
filtering was applied to $z_{0v\_ec}$ values, in particular, to avoid uncertainty in the bulk method due to non-neutral stability
conditions. However, using a filter that allows values from near-neutral conditions ($-0.1 < \frac{z}{L_{ec}} < 0.2$) rather than strictly
neutral, only ($\frac{z}{L_{ec}} = 0$), may introduce some uncertainty and variability to the $z_{0v\_ec}$ estimates. Furthermore, previous studies
have suggested that some assumptions of the bulk method, namely, constant momentum and heat flux values with height,
may not be valid during katabatic conditions with shallow wind maximums, which can develop frequently over sloped
glaciers (e.g. Denby and Smeets, 2000).



### 4.1.2 Snow Surfaces

Large differences in $z_{0v}$ between sites were also noted in this study for snow-covered surfaces. The annual persistence in roughness values observed over ice was also present in the snow surface values, with similar $z_{0v\_bloc}$ values returned for the same time each year when repeated at the same location. Where both *in situ* and remote values over snow surfaces were

available, agreement between $z_{0v\_ec}$ and the DEM-obtained roughness values varied substantially. In the case of CG16-2, which had a snow-covered surface throughout, the relatively large mean $z_{0v\_ec}$ value (2.4±16 mm) was substantially greater than $z_{0v\_bloc}$ and $z_{0v\_prof}$ (both 0.4 mm). This difference may be due to the temporal variance in roughness of a snow surface within a melt season (as observed in Fig. 6b), and the difference in observation time between the EC and LiDAR data. Images from the time lapse camera installed at CG16-2 (Fig. 14a-b) illustrate the variety in roughness conditions of the

snow surface at that site. Two periods were selected with visually apparent roughness differences and an adequate number of 30-minute $z_{0v\_ec}$ observations; a moderately smooth, melting snow surface (June 30th – July 3rd; 78 observations), and a rough, sun-cupped surface (Aug. 19th – 21st; 38 observations). Examining the $z_{0v\_ec}$ values, an order of magnitude difference was noted between the mean values for the moderately smooth (1.0±4.2 mm) and rough (9.6±21.7 mm) snow surfaces. In view of this short-term variability in snow roughness, the $z_{0v\_bloc}$ and $z_{0v\_prof}$ values, derived from LiDAR flights in April

and September, cannot be considered comparable to the $z_{0v\_ec}$ values from the summer. Imagery taken in the same location as the station site a few days after the April LiDAR flight (Fig. 14c) show a very smooth snow surface. With fresh snowfall in late August and September, a similar surface was likely present during the second flight, resulting in the small DEM-based values returned. Relatively large $z_{0v\_bloc}$ and $z_{0v\_prof}$ values were obtained for NG14 during the April LiDAR flights, possibly in response to a rough snow surface. Comparable *in situ* imagery of the site was not available for these periods,

however. The effect of the size of the roughness elements on a melting snow surface is discussed further in Sect. 4.2.

### 4.1.3 Wind Direction

Evidence of roughness length dependence on wind direction was observed in the 30-minute EC data at some locations, and in the rotated $z_{0v\_prof}$ values, also. The strongest dependence on wind direction in the $z_{0v\_ec}$ values was noted for the ablation zone of Conrad Glacier, at the location of CG15-2 and CG16-1. Elongated roughness features, including meltwater

channels, were present on the surface during these observations, with the orientation of their long axes pointing in a southeast to northwest direction (Fig. 12c). As the wind veered to the southwest, airflow became perpendicular to the faces of these features, likely resulting in increased form drag, which produced the larger roughness lengths observed. The rotated $z_{0v\_prof}$ values for the three stations in Conrad's ablation zone revealed an increase as wind direction approached a cross-glacier orientation. At NG14, the pronounced increase in roughness over the ice surface at 240° was likely due to a crevasse

field to the west of the station. This feature was not evident in the April values, suggesting snow cover had smoothed the surface in that region. Dependence of momentum roughness length on wind direction has been observed in several other glacier studies (e.g. Munro, 1989; Brock *et al*., 2006; Smith, 20014). Over all seasons and locations in this study, wind



direction was found to be within 45° of the mean slope angle for approximately 93% of the time. This persistent, katabatic downslope wind is a common feature in glacial boundary layers, and as a result, will substantially reduce the influence of surface roughness anisotropy on the variation in the effective roughness lengths and mean generated turbulence.

## 4.2 Performance of DEM-based $z_{0v}$ Estimation

The methods developed here for remotely estimating $z_{0v}$ were found to returning roughness length values within 1–2 mm (<< an order of magnitude) of those determined from *in situ* EC measurements, and were shown to respond to changes in surface cover from snow to ice. Using a DEM with a 1 x 1 m grid cell appears to resolve the length scales influencing $z_{0v}$ on the ice surfaces of this study. With a dense distribution of roughness elements (Fig. 12c), the previously mentioned effects of wake-interference and skimming of the airflow over the ablating ice may have reduced the influence of the smaller

roughness elements on $z_{0v}$, as noted in previous studies (e.g. Wieringa, 1993; Smeets *et al.*, 1999). During the April flights over Conrad Glacier, the DEM methods returned roughness values in line with previous observations over smooth, fresh or compacted snow surfaces (e.g. Brock *et al.*, 2006). Over rough, undulating snow surfaces, larger-scale features will have the dominant influence on roughness length (Fassnacht *et al.*, 2009), and are potentially resolvable in the utilised DEM, as may have been the case with the April values for NG14. Over smoother surfaces, however, it is likely that the roughness elements

influencing $z_{0v}$ are not resolvable with a 1 x 1 m DEM, making the usefulness of these methods over a melting snow surface uncertain (in addition to the temporal variation discussed in Sect. 4.1.2).

   The profile method developed here has been shown to return values in line with *in situ* estimates of the momentum roughness lengths without the need for the assumptions employed by the block method. The estimation of a similar cut-off wavelength at each station ($\lambda_0 = 35$ m) indicates a common length scale, above which, the surface features are no longer an

influence on roughness length. The $z_{0v\_prof}$ values did show a tendency towards overestimation, relative to the $z_{0v\_ec}$ values. In addition, the persistence between seasons in roughness length, noted in the $z_{0v\_ec}$ and $z_{0v\_block}$ values, was less evident in the $z_{0v\_prof}$ values, suggesting that the profile method is sensitive to changes in small scale features which may not have a substantial influence on the observed ($z_{0v\_ec}$) roughness values. The profile method also displayed sensitivity to the choice of DEM resolution, arising from substantial differences in the estimate of $s$ (Eq. 10) for different resolutions (>50% difference

between 1 x 1 m and 1 x 0.1 m resolutions).

   The block estimation method returned roughness length values that were smaller than those from the profile method, and more in line with mean $z_{0v\_ec}$, in general. The technique used in the $z_{0v\_bloc}$ method to calculate $s$ across overlapping block areas (as shown in Fig. 2 and Eq. 7) was developed in an effort to account for the shadowing of elements from airflow by upwind features. Rather than assuming that each feature above the mean surface has an additive influence on roughness

length, as done in the $z_{0v\_prof}$ method (below the cut-off wavelength) and other profile-based methods (e.g. Munro, 1989; Arnold and Rees, 2003), the relative height differences and potential sheltering influence of neighbouring features in the block are considered. On glacier surfaces, where elongated roughness features such as melt channels are common, the block




approach may also help account for the channelisation of air flow and the shadowing of the roughness element by the upwind continuation of the feature, which in turn, may reduce the effective roughness length. The response of the block method to this effect can be seen when the $F_{D\_local}$ estimates for the southerly (downslope) wind direction are compared with those for the westerly (cross-slope) wind direction (Fig. 15). Drag values estimated for the meltwater channels on the surface

are lower when air flow is close to parallel to these features, and higher when air flow is perpendicular to the channels. This effect may have led to the smaller $z_{0v\_bloc}$ values, relative to the $z_{0v\_prof}$ values. When implemented with an assumed turbulent footprint, the block method returned roughness length values in line with those calculated using a footprint model or from EC data (Table 5), indicating the potential for its use where turbulence observations are unavailable.

To apply the block approach, a number of additional assumptions were required, however. The choice of block size

corresponds to an assumption on the size of the dominant roughness elements influencing $z_{0v}$ on the glacier surface, and an assumption on the range of a feature's shadowing effect. The downwind shadowing generated by a feature will likely vary with wind speed, and this variation is not accounted for here. The optimal block size may vary between locations and wind regimes, and require tuning for application to other surfaces. Over the range of surfaces in this study, however, a 3 x 3 m block (applied to a 1 x 1 m DEM) was shown to be optimal, and to respond to changes in surface roughness due to snow and

ice cover. As a test of robustness, $z_{0v\_bloc}$ values were also estimated for a region of forest captured in the LiDAR data. This forest was located on a valley floor to the east of Conrad Glacier, and consisted of tall (~20 m), coniferous trees. The $z_{0v\_bloc}$ value for a 200 x 200 m subarea within this forest was 1.28 m. This value is in line with existing $z_{0v}$ measurements over coniferous forest (Wieringa, 1993). While the $z_{0v\_bloc}$ method (including the LiDAR data utilised) is not configured nor intended for use over forestry, this test indicates that its configuration (including selected block size) is responsive to a wide

range of roughness element sizes, beyond the scale of those encountered on the glacial surfaces of this study.

### 4.3 Scalar Roughness Relationships

Whilst displaying similar mean values over the entire dataset (0.05 mm for $z_{0t\_ec}$ and 0.11 mm for $z_{0q\_ec}$), the scalar roughness lengths differed substantially from each other when examined on a site-by-site basis. There was no evidence of a consistent ratio between $z_{0t\_ec}$ and $z_{0q\_ec}$, with their seasonal means ranging above and below each other by up to an order of

magnitude. Between the momentum and scalar roughness lengths, seasonal $z_{0t\_ec}$ displayed a more consistent relationship with $z_{0v\_ec}$, being approximately one and a half orders of magnitude smaller than $z_{0v\_ec}$ in most cases. This relation did not hold for NG14 and CG16-2, however, and between $z_{0v\_ec}$ and $z_{0q\_ec}$, there was no persistent ratio. Calanca (2001) observed $z_{0t}$ to be a function of the temperature gradient between the air and a melting ice surface, while Park *et al.* (2010) found a relation between relative humidity at 2 m height and $z_{0q}$. In this study, variation in the scalar roughness lengths was

compared with fluctuations in air temperature gradient and relative humidity, but no dependent relationship was evident. The surface renewal model of Andreas (1987), where the ratio of momentum to scalar roughness was expressed as a function of $R_*$, showed relatively good performance, particularly for seasonal values of $z_{0t}$. If momentum roughness length values have





been obtained for a given surface (through remote or *in situ* methods), this model appears to be the best available method for estimating the scalar values.

## 5 Conclusions

Over three melt seasons, *in situ* and remote methods were implemented to determine the momentum and scalar roughness lengths on the surface of two glaciers in the Purcell Mountains of British Columbia, Canada. EC sensors were employed to obtain continuous *in situ* measurements throughout each melt season, while LiDAR-derived DEMs were utilised in the development of two remote estimation techniques. Seasonal mean momentum roughness length values, estimated from eddy covariance observations at each location, ranged from 0.7–4.5 mm for ice surfaces, and 0.5–2.4 mm for snow surfaces. For representative turbulent flux modelling, this study suggests that site-specific $z_{0v}$ values are necessary, particularly in the case of distributed glacier models. From year-to-year, $z_{0v}$ values were noted to remain relatively consistent at a given location (<0.2 mm difference between seasonal mean values). Within a melt season, continuous EC observations and camera imagery noted greater temporal variation in roughness for snow surfaces than for ice. These findings indicate that site-specific $z_{0v}$ values on an ice surface may be valid to implement over multiple melt seasons, while over snow surfaces, the utilised roughness values require intraseasonal updating. Wind direction was also noted to affect $z_{0v}$ variability where elongated features such as melt channels dominated the surface topography. Persistence in wind direction on sloped glacier surfaces, however, reduces the influence of this variability.

Observations of the scalar roughness lengths differed substantially from the corresponding momentum values, showing considerable variation between location and season, and little agreement with fixed ratios commonly assumed with $z_{0v}$. In general, the Andreas (1987) surface renewal method showed agreement with the observed ratios between EC-derived scalar and momentum roughness lengths, and would seem to be the appropriate method to implement where continuous EC observations are not available, but site-specific $z_{0v}$ values have been established.

The DEM-based methods described in this study were shown to perform well over most surfaces, differentiating between ice and snow cover, and returning momentum roughness values that were within 1–2 mm (<< an order of magnitude) of EC-derived values for the corresponding footprints. The features dominating glacier surface roughness, particularly over ice, were found to be of a scale that was resolvable using a 1 x 1 m DEM, and persistent enough for a DEM-based roughness estimate to be usable over an extend period of time. This may allow for the potential upscaling of these methods with high resolution satellite imagery, greatly expanding the number of glaciers for which roughness length estimates could be obtained. Over melting snow surfaces, the validity time of a retrieved DEM is reduced due to the discussed temporal variability in roughness, and as a result, the estimated roughness lengths may quickly become unrepresentative. In addition, the roughness features observed to develop on melting snow in this study may not be resolvable using a 1 x 1 m DEM, and further testing over snow, with simultaneous *in situ* and remote observations, would be useful.



*Author contributions*. N. Fitzpatrick prepared and executed the on-glacier field campaign, conducted the data analysis, and proposed the roughness estimation methods presented in the chapter, as well as writing the text. V. Radić helped develop these methods and the associated sensitivity testing, supervised the analysis, and provided financial and field support. B. Menounos instigated the LiDAR imaging flights, and provided the resulting digital elevation models to the study, in addition to logistical and minor financial support to the field campaign.

*Competing interests*. The authors declare that they have no conflict of interest.

*Acknowledgements*. Funding support for this study was provided through the Natural Sciences and Engineering Research Council (NSERC) of Canada (Discovery grants to V. Radic and B. Menounos). The eddy covariance system and meteorological equipment were supported by NSERC Research Tools and Instruments grant and Canada Foundation for Innovation grant (to V. Radic). The authors wish to thank Ben Pelto for his assistance in the field and with the provision of the DEMs, Steve Conger and Tannis Dakin for logistical support, Zoran Nesic for assistance with station design and sensor calibration, and Jörn Unger for his work on the quadpod construction.

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





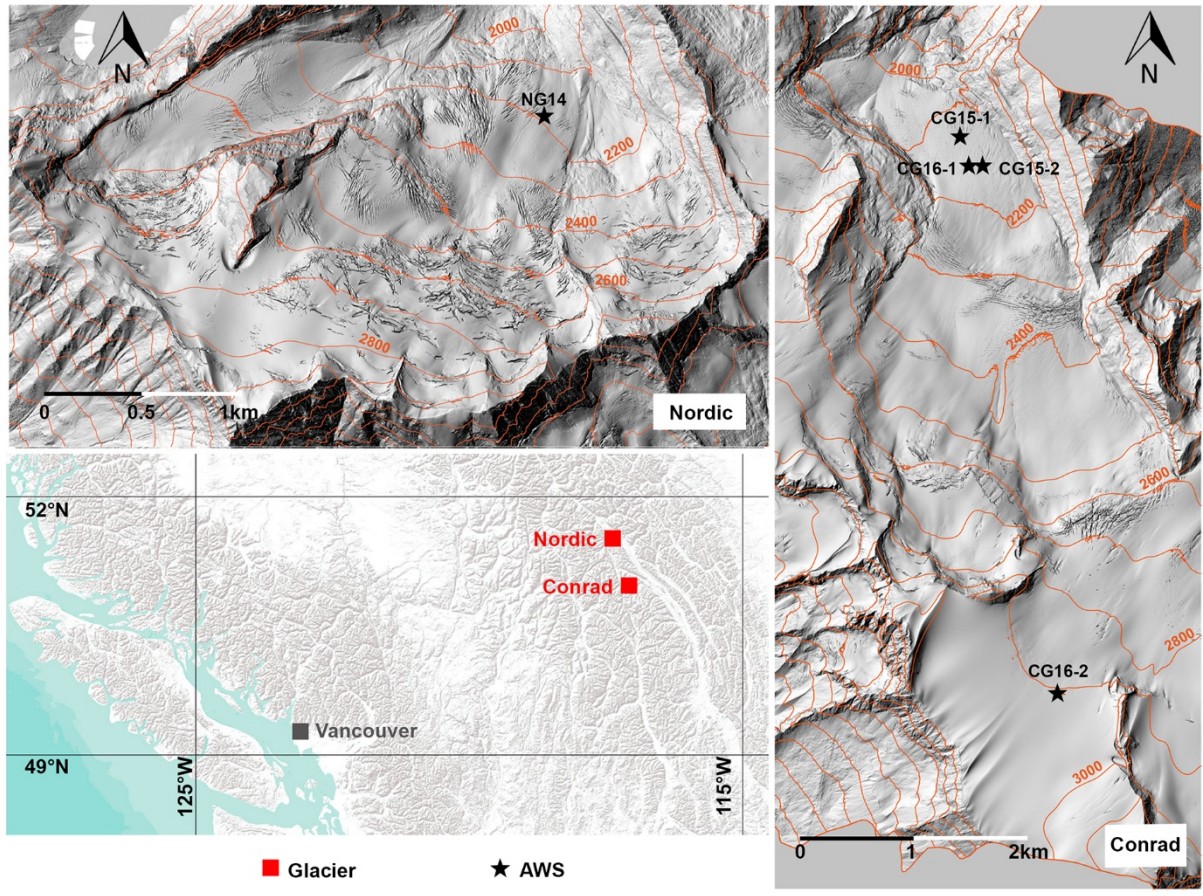

Figure 1. Location of the study glaciers and the stations installed during the 2014–2016 melt seasons.





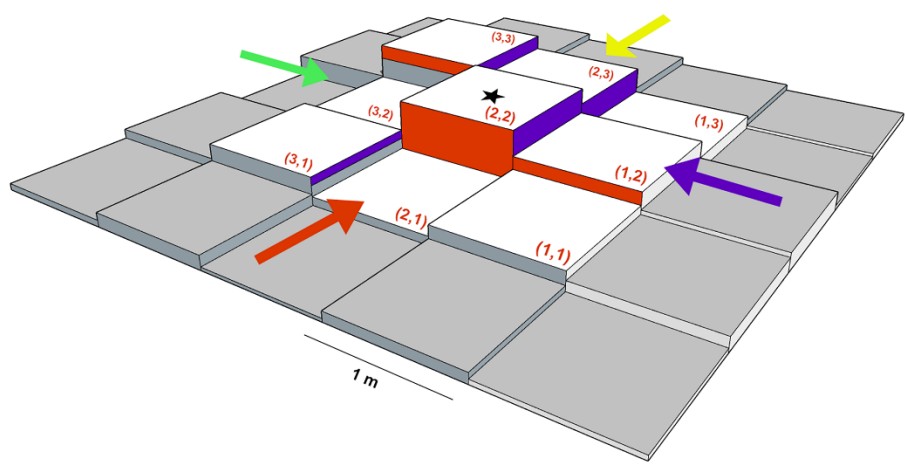

**Figure 2. DEM-based block method for estimating the local drag generated by roughness elements on the surface. The total surface area that is perpendicular and 'visible' to the direction of air flow (matching-coloured face area and arrows) is assigned to**
5 $s_b$ **(Eq. 7). The displayed grid cell indices are for airflow in the direction of the red arrow. A $F_{D\_local}$ value is estimated for the four cardinal wind directions, with the values assigned to the central grid cell of the block (starred). The block is then moved by one grid cell at a time, and the process repeated over the DEM.**



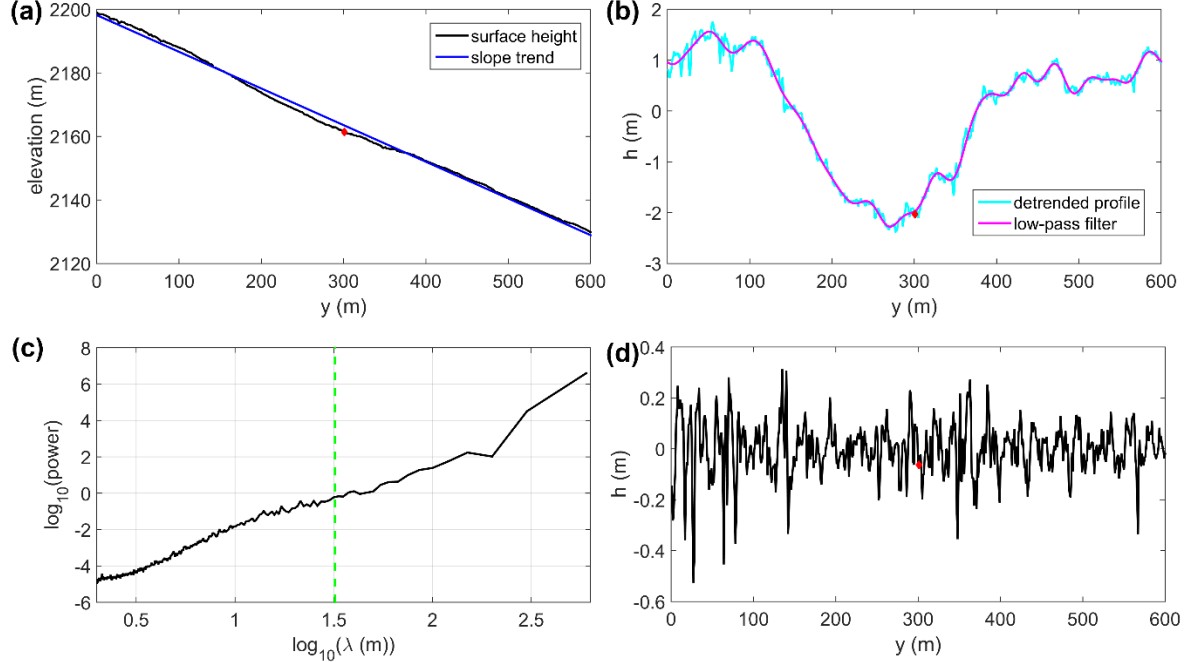

**Figure 3. (a) Surface height profile from the September 2016 DEM centred on CG16-1 (red diamond) and a fitted linear trend; (b) detrended profile and low-pass filter according to cut-off wavelength of $\lambda_0$; (c) log-log power spectrum of the mean detrended profile, with large scale wavelengths greater than $\lambda_0$ (green dashed line) used in the low-pass filtering; (d) filtered profile used in the calculation of momentum roughness length.**

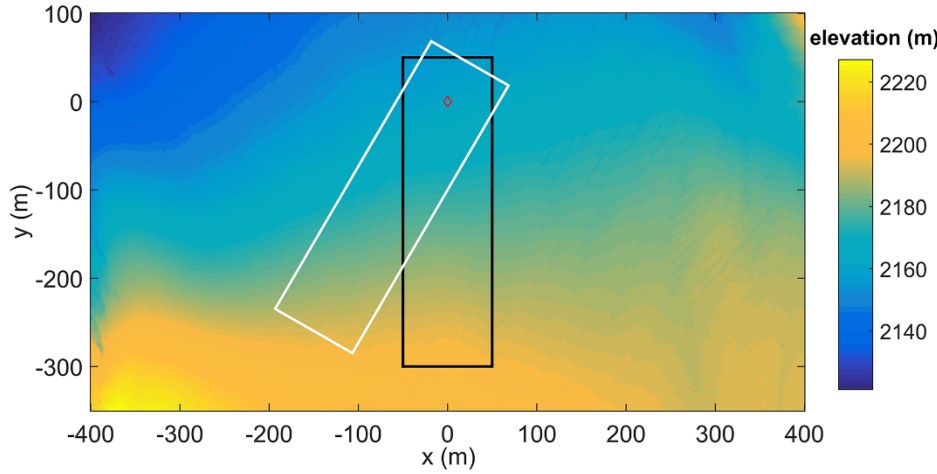

**Figure 4. Example of the rotation applied to a DEM patch selected around a station location (red diamond), with the original orientation outlined in black, and a rotated patch, turned 30° clockwise, outlined in white.**





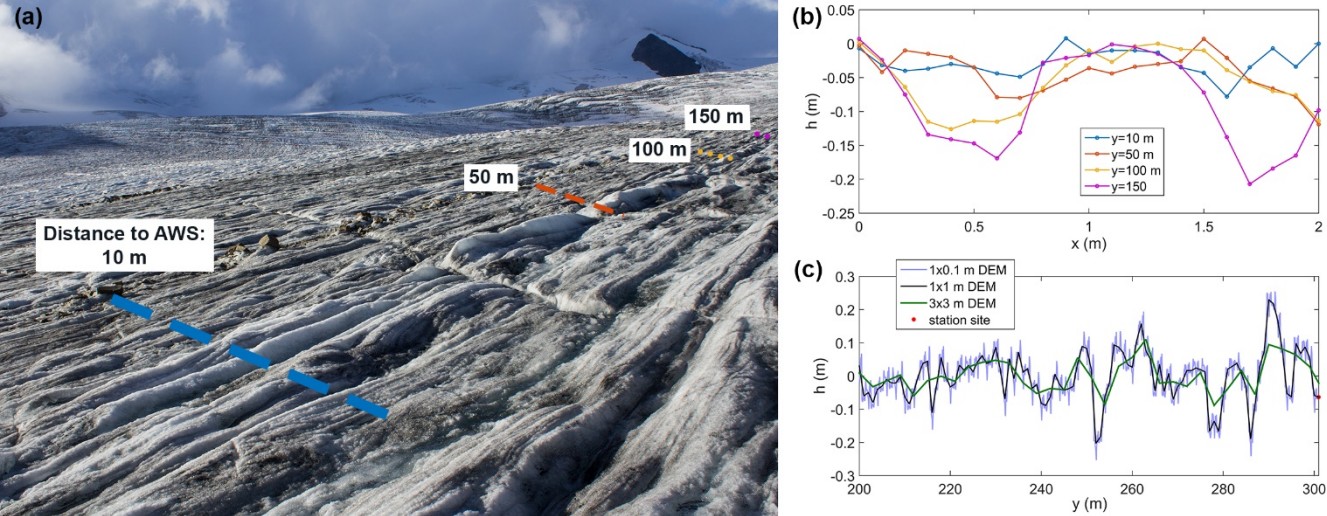

**Figure 5. (a & b) Microtopography profiles taken upwind of CG16-1 at the end of the 2016 melt season. Profiles were 2 m in width and taken perpendicular to the downslope direction. The locations of the profiles marked in (a) are representative rather than exact. (c) Examples of the filtered height profiles, as derived from the three DEM resolutions used in the $z_{0v\_prof}$ sensitivity test.**

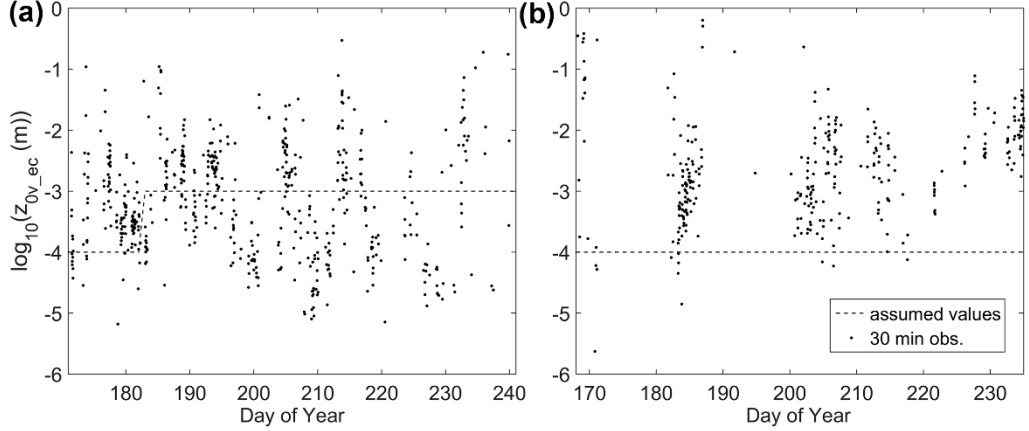

**Figure 6. 30-minute $z_{0v\_ec}$ values as observed at (a) CG16-1 and (b) CG16-2. The dash line represents the commonly assumed $z_{0v}$ values of 1 mm and 0.1 mm for ice and snow, respectively. At CG16-1, the surface transitioned to bare ice on day of year (DOY)**
10      **183.**



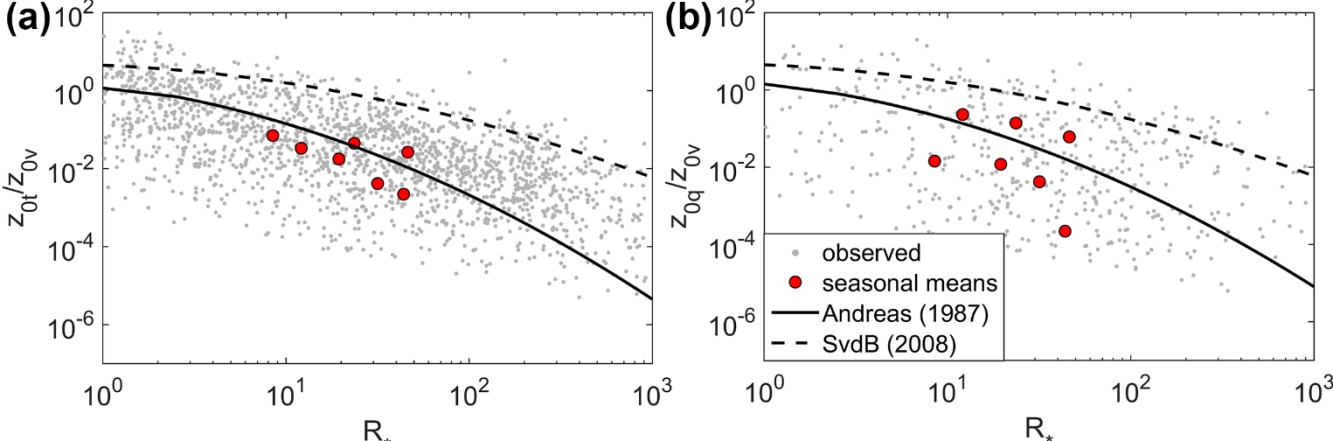

**Figure 7. Performance of the surface renewal models of Andreas (1987) and Smeets and van den Broeke (2008) for estimating the ratio of (a) $z_{0t}$ and (b) $z_{0q}$ to $z_{0v}$. The filtered 30-minute (grey) and seasonal mean (red) ratios of the EC-derived roughness lengths and $R_*$ values are shown for all seasons and EC sensors.**

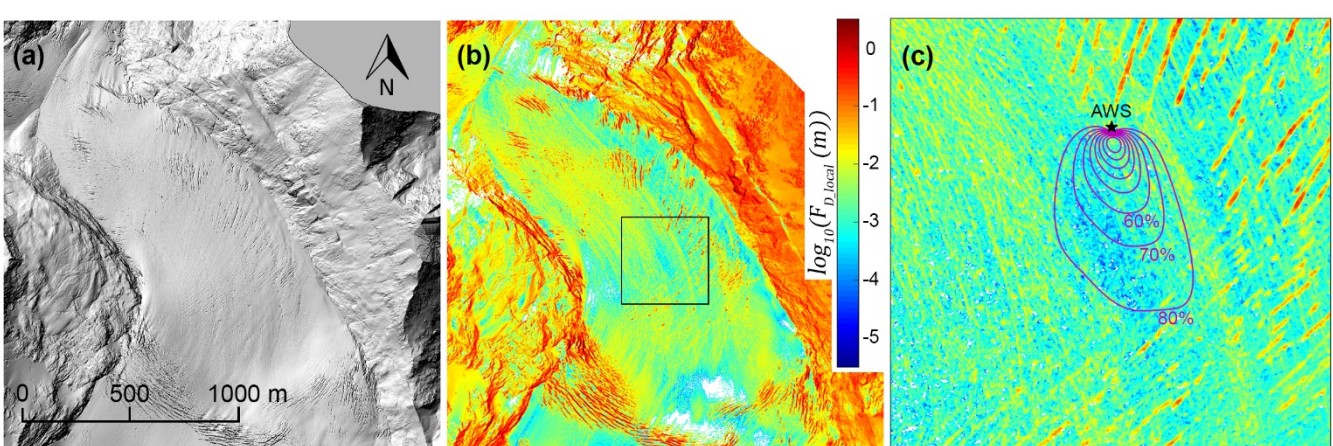

**Figure 8. Example from CG16-1 of the steps taken to estimate $z_{0v\_bloc}$ from LiDAR data: (a) 2,000 x 2,000 m subarea extracted
10  from the 1 x 1 m DEM, centred on an AWS; (b) localised drag values ($F_{D\_local}$) calculated for each grid cell; (c) the flux footprint for the corresponding EC data, shown as percentage of crosswind integrated flux contribution (purple contours), overlaid over the $F_{D\_local}$ map (400 x 400 m square area expanded from (B) for display purposes).**





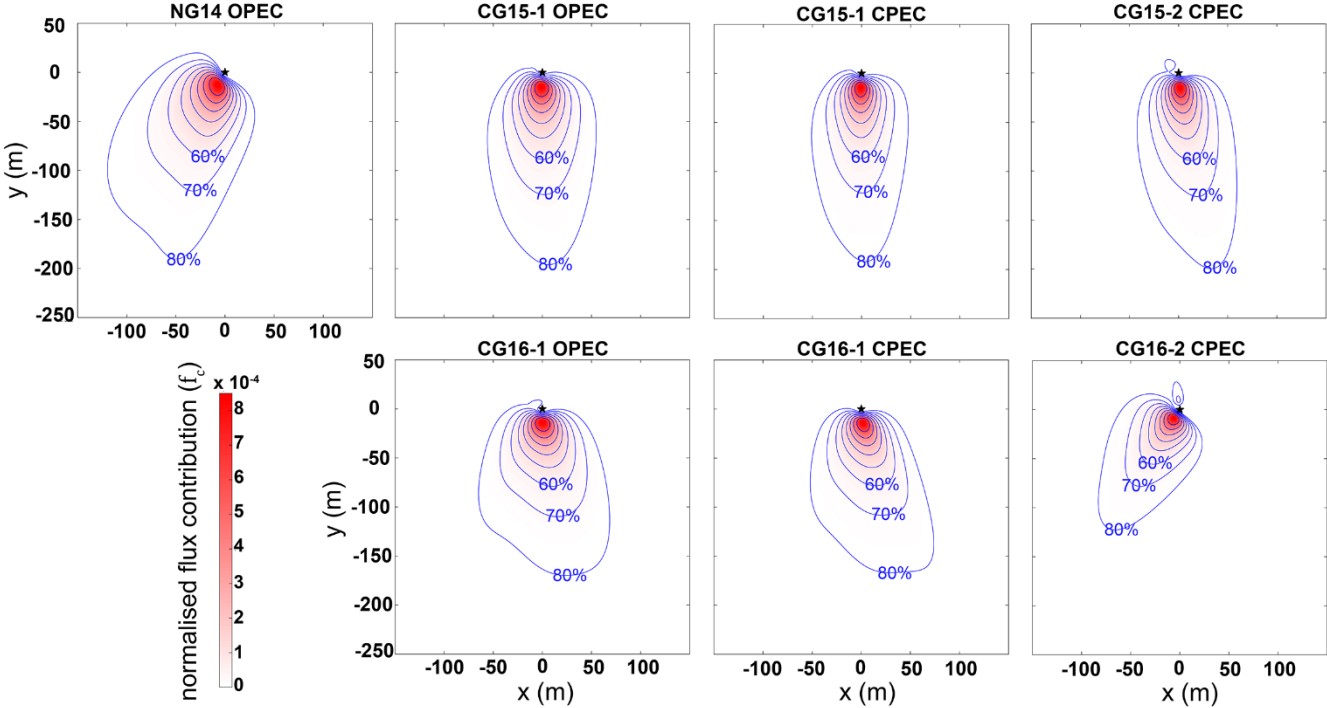

**Figure 9. Flux footprint maps for each EC system deployed during the study, including percentage of crosswind integrated flux contribution (purple contours). Distances are in metres east (x) and north (y) of the AWS (black star). Maps were produced following the methods of Kljun _et al._ (2015).**





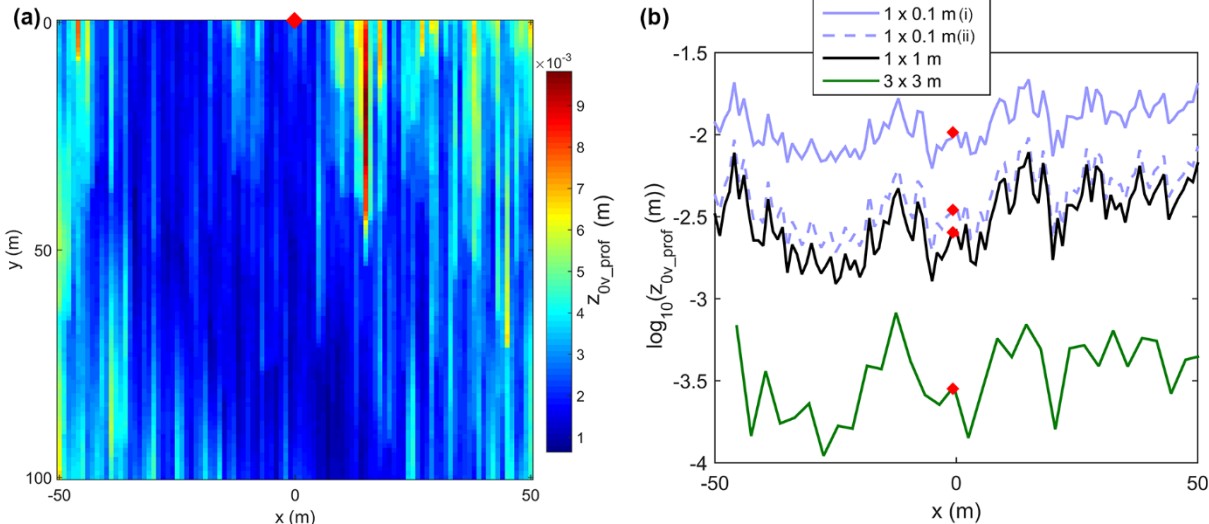

**Figure 10. (a)** $z_{0v\_prof}$ **values estimated for each grid cell in a 101 x 101 m area upwind from CG16-1 (red diamond) from the September 2016 DEM of Conrad Glacier. (b)** $z_{0v\_prof}$ **values derived for the downslope profiles at CG16-1 (x = 0) and for the grid cells 50 m to the east and west of the station from the original DEM (1 x 1 m), and from the higher (1 x 0.1 m) and lower (3 x 3 m) resolution DEMs constructed for sensitivity testing. The 1 x 0.1 m (i) values are from the initial high resolution DEM used in the sensitivity test, while the DEM used for 1 x 0.1 m (ii) had the amplitude of the synthetic microtopography profiles reduced by a factor of 10.Table 1. Locations and dates of operation of the automatic weather stations used in this study.**

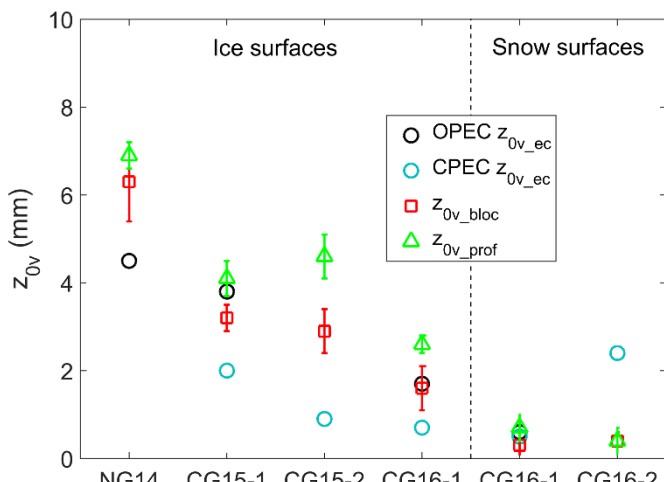

**Figure 11. Comparison of geometric mean OPEC and CPEC momentum roughness length observations with** $z_{0v\_bloc}$ **and** $z_{0v\_prof}$ **estimates from the remote methods. Values are separated into ice and snow surface types. Error bars represent the calculated uncertainty in the** $z_{0v\_bloc}$ **method, and σ of the** $z_{0v\_prof}$ **values for an 101 x 101 m upwind patch. The standard deviation on each of the mean** $z_{0v\_ec}$ **values (see Table 4) extends beyond the y-axis range.**





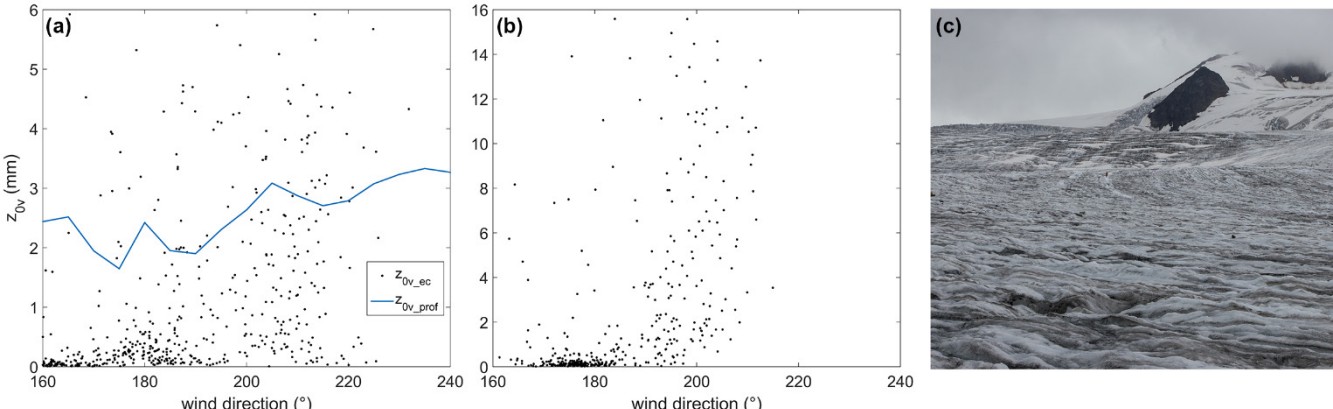

**Figure 12. The dependence of momentum roughness length values on wind direction for (a) $z_{0v\_ec}$ (filtered 30-minute values) and $z_{0v\_prof}$ at CG16-1, and (b) $z_{0v\_ec}$ at CG15-2 (LiDAR data was not available for CG15-2 during this period). (c) Elongated roughness features on Conrad glacier, looking south from CG15-1 in July 2015. The meltwater channels had approximate dimensions of 0.5–1 m in width and 0.1–0.2 m in depth, with substantial variability.**

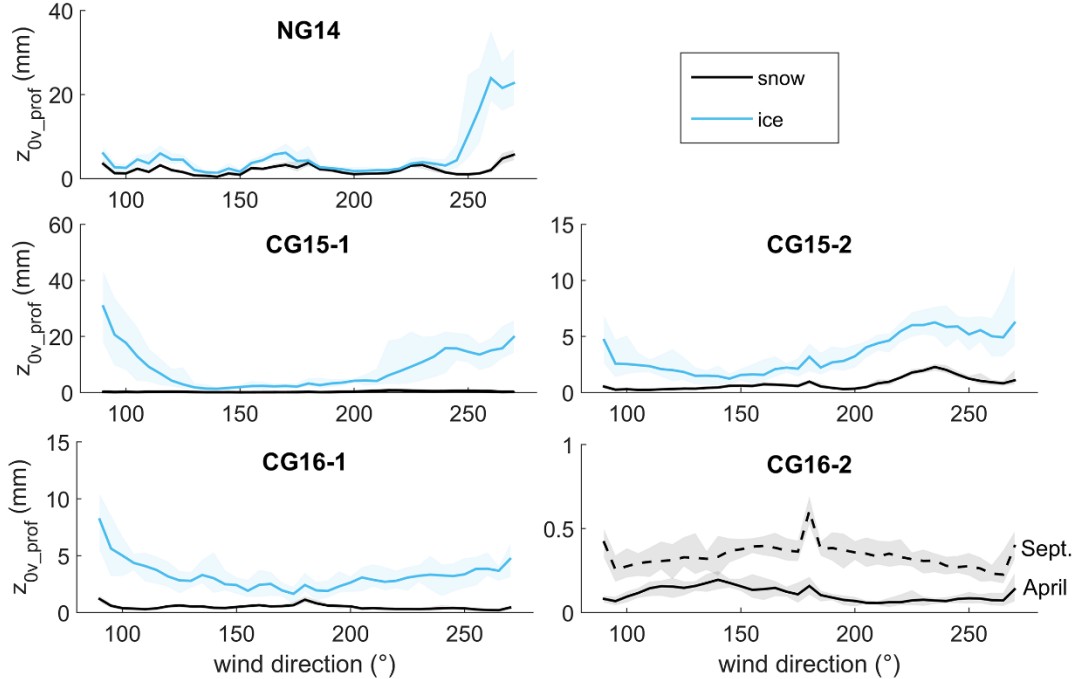

**Figure 13. Roughness values from the rotated $z_{0v\_prof}$ method for 5° increments in wind direction between 90° and 270° for an April (snow) and September (ice) surface at each station (snow surface present at CG16-2 for both periods). Shaded area represents the range of roughness lengths estimated for the five profiles either side of the station profile (11 profiles).**



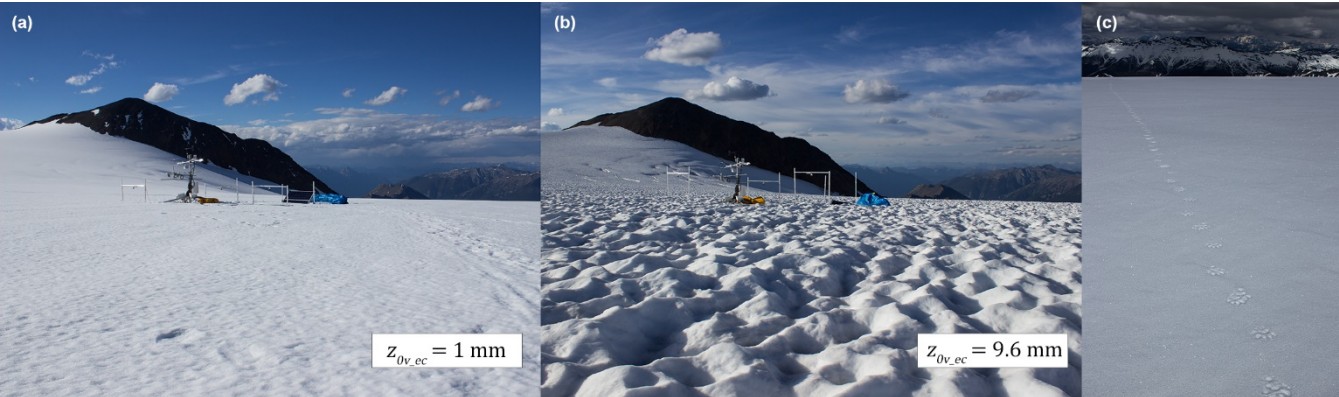

**Figure 14.** Observed snow surface roughness variations at CG16-2 from camera imagery for (a) June $30^{th}$ – July $3^{rd}$ ($z_{0v\_ec}$ = 1.0±4.2 mm), and (b) Aug. $19^{th}$ – $21^{st}$ ($z_{0v\_ec}$ = 9.6±21.7 mm). For scale, the upper crossarm of the AWS is at a height of 1.9 m. (c) Smooth snow surface observed at the location of CG16-2 on April $26^{th}$, 2016 (wolverine tracks for scale).

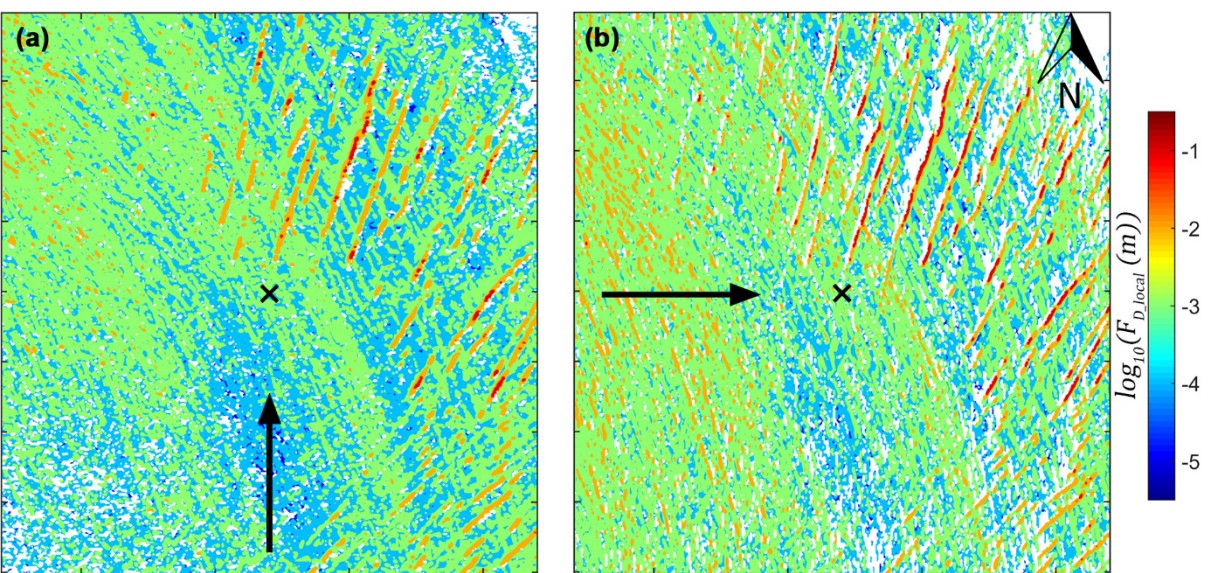

**Figure 15.** Comparison of the $F_{D\_local}$ values from the block method for (A) southerly (downslope) and (B) westerly (cross-slope) wind direction at CG16-1 in September 2016. Air flow (black arrow), and AWS location (black cross) are also identified.





**Table 1. Locations and dates of operation of the automatic weather stations used in this study.**

| Station | NG14 | CG15-1 | CG15-2 | CG16-1 | CG16-2 |
|---|---|---|---|---|---|
| **Glacier** | Nordic | Conrad | Conrad | Conrad | Conrad |
| **Location** | 51.43434°N | 50.82486°N | 50.82306°N | 50.82303°N | 50.78219°N |
| | 117.69973°W | 116.92247°W | 116.92128°W | 116.91992°W | 116.91197°W |
| **Zone** | ablation | ablation | ablation | ablation | accum. |
| **Elevation** | 2208 m | 2138 m | 2163 m | 2164 m | 2909 m |
| **Deployed** | 12/07/2014 | 15/07/2015 | 16/07/2015 | 19/06/2016 | 16/06/2016 |
| **Removed** | 28/08/2014 | 05/09/2015 | 07/09/2015 | 28/08/2016 | 22/08/2016 |

**Table 2. Instrument list for each deployed station, including sensor accuracy and height of installation.**

| Variable | Sensor | Accuracy | NG14 | CG15-1 | CG15-2 | CG16-1 | CG16-2 |
|---|---|---|---|---|---|---|---|
| Wind speed/direction | Young 05103ap Wind Monitor | ±0.3 m s$^{-1}$ | ● | ● | ● | ● | ● |
| Air temperature/humidity | Rotronic HC2 Probe | ±0.1°C / 0.8% | ● | ● | ● | ● | ● |
| Air temperature/humidity | Aspirated Rotronic HC2 Probe | ±0.1°C / 0.8% | - | - | - | ● | ● |
| Atmospheric Pressure | Vaisala PTB110 | ±0.3 hPa | ● | ● | ● | ● | ● |
| Precipitation | Texas Elec. Tipping Bucket Gauge | ±1% (up to 10 mm hr$^{-1}$) | ● | ● | ● | ● | ● |
| Radiation fluxes | Kipp & Zonen CNR4 | 10–20 W m$^{-2}$ (pyranometer)<br>5–15 W m$^{-2}$ (pyrgeometer) | ● | ● | ● | ● | ● |
| Turbulent fluxes:<br>  water vapour<br>  3D wind (u,v,w)<br>  sonic temp | OPEC System<br>  CSI IRGASON<br>  CSI IRGASON<br>  CSI IRGASON | <br>3.5 x 10$^{-3}$ g m$^{-3}$<br>1 mm s$^{-1}$<br>±0.025°C | ● | ● | - | ● | - |
| Turbulent fluxes:<br>  water vapour<br>  3Dwind (u,v,w)<br>  sonic temp | CPEC System<br>  LI-7200<br>  Gill R3-50<br>  Gill R3-50 | <br>±2%<br><1% RMS<br>±0.1°C | - | ● | ● | ● | ● |
| Ground heat flux | Thermistor Array (self) | ±0.1°C | ● | ● | ● | ● | ● |
| Surface height | CSI SR50A Sonic Ranger | ±0.01 m | 1 | 3 | 3 | 3 | 3 |
| Surface temp | Apogee SI-111 | ±0.2°C | - | 1 | 1 | 2 | 2 |
| Station tilt | Turck Inclinometer | ±0.5° | ● | ● | ● | ● | ● |
| Data storage | CSI CR3000 Logger | - | ● | ● | ● | ● | ● |
| Site/Surface Conditions | Time Lapse Camera (self) | - | ● | ● | ● | ● | ● |
| $z$ (m) | - | - | 2.0 | 2.0 | 2.0 | 1.9 | 1.9 |
| $z_u$ (m) | - | - | 2.6 | 2.5 | 2.6 | 2.6 | 2.4 |



**Table 3. Dates of LiDAR flights over the two study glaciers from 2014 to 2016. *For the September 12th 2015 flight over Conrad Glacier, only the accumulation zone was adequately captured.**

|  | Nordic Glacier | | Conrad Glacier | |
|---|---|---|---|---|
|  | **Spring** | **Autumn** | **Spring** | **Autumn** |
| **2014** | July 10th | Sept 11th | - | Sept 11th |
| **2015** | April 19th | Sept 11th | April 20th | Sept 12th * |
| **2016** | April 17th | Sept 12th | April 17th | Sept 12th |



**Table 4. Seasonal geometric means of the EC-derived roughness length values (±σ) from the open and closed path systems for each station site. $z_{0v\_ec}$ values for periods with a snow-covered surface are underlined. The number of 30-minute periods available for roughness estimation (after filtering) is presented in square brackets.**

| (mm) | NG14 OPEC | CG15_1 OPEC | CG15_1 CPEC | CG15_2 CPEC | CG16_1 OPEC | CG16_1 CPEC | CG16_2 CPEC |
|---|---|---|---|---|---|---|---|
| $z_{0v}$ | 4.5±28.8 [93] 0.46±3 [16] | 3.8±31.7 [206] | 2.0±19.2 [281] | 0.9±7.4 [417] | 1.7±11.7 [308] 0.62±5.1 [114] | 0.7±6.4 [429] 0.51±2.3 [138] | 2.4±16 [312] |
| $z_{0t}$ | 0.01±0.1 [77] | 0.01±0.88 [181] | 0.09±0.81 [270] | 0.03±0.28 [390] | 0.03±0.23 [396] | 0.05±0.29 [546] | 0.01±0.07 [247] |
| $z_{0q}$ | 0.001±0.008 [16] | 0.23±1.5 [43] | 0.28±1.9 [17] | 0.21±3.1 [74] | 0.02±0.28 [194] | 0.01±0.19 [186] | 0.01±0.1 [38] |



**Table 5.** Momentum roughness length values (in mm) for each station estimated using remote methods ($z_{0v\_bloc}$ and $z_{0v\_prof}$) from the LiDAR-derived DEMs. The roughness values for the prevailing downslope southerly wind direction are shown here. $f_{c\_100}$ represents values for an assumed 101 x 101 m upwind footprint where $F_{D\_local}$ values are given equal weighting. The uncertainty values from error propagation are shown for $z_{0v\_bloc}$, while for $z_{0v\_prof}$, $\pm\sigma$ of the roughness values for the 101 x 101 m upwind patch is presented.

| | NG14 | | CG15-1 | | CG15-2 | | CG16-1 | | CG16-2 | |
|---|---|---|---|---|---|---|---|---|---|---|
| $z_{0v\_bloc}$ | April | Sept | April | Sept | April | Sept | April | Sept | April | Sept |
| **2014** | - | 6.3±0.9 | - | 2.5±0.1 | - | 2.5±0.5 | - | 1.6±0.4 | - | 0.5±0.2 |
| **2015** | 2.0±0.2 | 5.0±0.1 | 0.3±0.2 | - | 0.5±0.2 | - | 0.3±0.2 | - | 0.3±0.1 | 0.4±0.2 |
| **2016** | 2.5±0.1 | 4.0±0.4 | 0.6±0.3 | 4.0±0.4 | 0.8±0.2 | 3.2±0.5 | 0.3±0.1 | 1.6±0.5 | 0.4±0.1 | 0.4±0.1 |
| | | | | | | | | | | |
| $f_{c\_100}$ | 2.0±0.2 | 3.2±0.4 | 0.5±0.2 | 4.2±0.2 | 0.6±0.2 | 2.1±0.5 | 0.2±0.1 | 0.9±0.4 | 0.2±0.1 | 0.2±0.1 |
| | | | | | | | | | | |
| $z_{0v\_prof}$ | | | | | | | | | | |
| **2014** | - | 6.9±0.3 | - | 2.6±0.2 | - | 2.0±0.3 | - | 2.1±0.2 | - | 0.4±0.02 |
| **2015** | 4.6±0.4 | 4.2±0.4 | 0.2±0.04 | - | 0.5±0.02 | - | 0.9±0.03 | - | 0.1±0.01 | 0.2±0.04 |
| **2016** | 3.6±0.2 | 5.6±0.1 | 0.6±0.1 | 5.6±0.5 | 1.7±0.1 | 7.1±0.6 | 0.7±0.03 | 2.6±0.2 | 0.1±0.02 | 0.6±0.04 |



**Table 6. Comparison of momentum roughness length values (in mm) for each station, as observed from the EC systems ($z_{0v\_ec}$), and as estimated using the DEM-based methods ($z_{0v\_bloc}$ and $z_{0v\_prof}$). \*For years where LiDAR data was not available from the same year a station was in place, the averages of the roughness estimates from the two other years were utilised for evaluation.**

| | NG14 OPEC | CG15-1 OPEC | CG15-1 CPEC | CG15-2 CPEC | CG16-1 OPEC | | CG16-1 CPEC | | CG16-2 CPEC |
|---|---|---|---|---|---|---|---|---|---|
| | ice | ice | ice | ice | snow | ice | snow | ice | snow |
| $z_{0v\_ec}$ | 4.5 | 3.8 | 2 | 0.9 | 0.6 | 1.7 | 0.5 | 0.7 | 2.4 |
| $z_{0v\_bloc}$ | 6.3 | 3.2* | 3.2* | 2.9* | 0.3 | 1.6 | 0.3 | 1.6 | 0.4 |
| $z_{0v\_prof}$ | 6.9 | 4.1* | 4.1* | 4.6* | 0.7 | 2.6 | 0.7 | 2.6 | 0.4 |