# Peer review of "A multi-season investigation of glacier surface roughness lengths through in situ and remote observation"

_The Cryosphere, 2018_

## Referee Comment (RC1) · Smith (Referee) · 11 Dec 2018

General comments

This well-written paper provides an original and significant contribution to the study of roughness lengths on mountain glaciers. It was extremely interesting to read and it provides new insights into roughness length estimation through quantitative comparison of extensive and rigorously collected data sets. The analysis does much to close the gap between eddy covariance and topography derived aerodynamic roughness estimates (section 3.2.1 was very interesting in that regard with some highly citeable findings).

[Figure]

The paper itself is the correct length, contains clear figures and is very easy to read. I believe it to be publishable in almost its present form, though I have a few minor points listed below. These are generally points of clarification. However, I did find the synthetic DEM approach to investigate the effect of smaller scale roughness on roughness length to be rather unconvincing. I do not feel that the paper needs this in any case.

Specific comments

(1) Other studies estimating z0 on mountain glaciers

The introduction is clear and concise, presenting a clear rationale for the study while acknowledging previous work. In general, this paper adequately cites and gives ample credit to other studies. However, I can identify a few specific papers that contain findings of some relevance to the study here. In the majority, they add greater weight to the arguments and findings presented. While I agree with the comment (P3, L6) that "Similar studies on mountain glaciers are very rare", there are a few examples for Himalayan (debris-covered) glaciers that might be worth considering (Quincey et al., 2017; Miles et al., 2017).

Miles, E.S., Steiner, J.F. and Brun, F., 2017. Highly variable aerodynamic roughness length (z0) for a hummocky debris‐covered glacier. Journal of Geophysical Research: Atmospheres, 122(16), pp.8447-8466.

Quincey, D., Smith, M., Rounce, D., Ross, A., King, O. and Watson, C., 2017. Evaluating morphological estimates of the aerodynamic roughness of debris covered glacier ice. Earth Surface Processes and Landforms, 42(15), pp.2541-2553.

(2) Block estimation method

The paper presents an interesting DEM-based method for estimating z0 from micro-topography. Overall, this appears rigorous and correct, but I do have a few minor questions for clarification regarding the implementation of the method.

- Equation (7) would benefit from some clarification. It took me a while to understand that, but I was simply confused by the definition of 'lines' and 'rows'. Aside from that, it makes sense – Figure 2 is extremely helpful. How does this work when the blocks have a larger cell dimension? Do you assume sheltering across the whole 11 m?

- How were edge effects of using a moving window dealt with? Was it extrapolated or padded with zeros?

- Is the result of equation 8 not really a localised $z_0$? Why is it given different notation and called a drag value?

- What are these "assumed footprint areas" (P9L18)? It sounds interesting, but I'd like to see more details (mentioned again P18, L7). Actually, the finding on P12L30 that equally weighted cells gave similar results is very interesting and helpful for studies going forwards. Is the assumed footprint related to this?

(3) Profile estimation method

I found this method to be very interesting. In fact, it differs markedly from the conventional 'profile' approach (e.g. Munro, 1989) in that wind parallel profiles are taken, rather than perpendicular profiles. This works well when dealing with topographic data at the scale available here (1 x 1 m). I find myself agreeing with the authors' approach here, but I think the difference between this and previous work could be flagged. Once again, I have a few points for clarification:

- The cut-off wavelength of 35 m seems to be important, but looking at Figure 3c, I cannot quite see why this was decided upon.

- Equation 10 shows that the absolute value of the elevation difference between cells is used in the estimate of s. Is this correct? Is it not more appropriate to consider the differences facing the wind (i.e. only considering positive differences, rather than turning negative differences into positive values)?

- As with the block method, how were edge effects dealt with?

Munro, D.S., 1989. Surface roughness and bulk heat transfer on a glacier: comparison with eddy correlation. Journal of Glaciology, 35(121), pp.343-348.

(4) DEM scale sensitivity

I find the creation of the synthetic DEM at a finer scale than the LiDAR data to be the least convincing aspect of the paper. In my opinion, it detracts from the key messages of the paper and is better off being removed entirely. It would make a useful discussion point for further work, but I am not convinced that the data support this analysis.

Technical corrections

P4L22: data was -> data were

Table 2 – are z and zu defined in the text anywhere?

P10L30 – should the second instance of 'parallel' read 'perpendicular'?

P10L33 –re-word 'demeaned'!

Section 3.2.3 – to confirm, is this the 1 x 1 m data?

P16L32 – note typo in citation
* * *

---

## Referee Comment (RC2) · Litt (Referee) · 14 Dec 2018

General comments

The work described in this paper addresses a key challenge: the characterization of the surface roughness lengths is highly relevant for glacier and snow energy balance modelling. It is totally in the scope of the Journal. The paper is well structured, nicely written, and proposes clear figures and tables which facilitate the overview of the instrumental setting and, to be more specific, of the different surface's temporal and spatial scales implied.

[Figure]

A new approach, the block-method, and the profile method, an enhanced/modified version of a previously published method, are proposed to characterize the surface roughness parameters, and these are clearly described. These are compared to the roughness lengths derived by inverting the equation describing the well-known bulk-aerodynamic approach, in which high frequency eddy covariance measurements are used to calculate the instantaneous fluxes. Footprint calculations and assessment are included which gives the results a strong basis, and all these methods have been implemented really carefully, using on-field data which are - from the paper perspective- of high-quality.

This paper should really be published, since the work is highly relevant for future investigation. Though, a few important key issues, pointing towards the basis of the above mentioned methods, are only briefly mentioned, thought they might play a fundamental role in the divergences observed in the results from the different methods. I don't think any key calculations would have to be remade, but at least a stronger discussion of these issues should be included, and an assessment of the impact that may have on the observed results, in order for readers and the community not to blindly follow these.

Please find these comments below.

The eddy covariance derived surface roughness and the assumptions of the similarity theory. All along the paper, it is assumed that the 30 min-averaged values of $u^*$, $T^*$ and $q^*$, are collected during flow conditions which are representative of the assumptions of the similarity theory, a necessary condition for the applicability of the bulk-formulation. This is guaranteed through the careful filtering and selection of the high frequency data, for various criteria (i.e, minimum wind speed, specific flow direction, near-neutral stability conditions, etc., section 2.4, lines 20 to 28). This ensures the scaling of the bulk formulation is valid only if the state of the turbulent flow is driven only by its interaction with the surface, in other words if turbulence is generated only by the interaction of the flow with the surface, and depends only on the stability of the surface layer. One key filter that ensures this conditions to be fulfilled is the one for stationarity of the

flow and fully developed turbulence. Thought, even in these cases it is possible that additional turbulence is transported through by large eddies originating from outside layers – related processes, or being transported away, so that the actual measured u*, T*, q* do not scale with the mean flow surface properties (Hogstrom, 2002, Thomas and Foken 2005, Barthlott et al 2007, Litt et al., 2015), even under neutral stability conditions. In such cases it is most likely that the roughness provided by the High Frequency fluxes will not be representative of the surface characteristics, and so that the roughness inferred out from the actual surface state (DEM block, profile methods) do not reproduce the actual fluxes when using the bulk formulation. If the filtering retains only specific meteorological conditions for which the turbulent flow develops in a certain way, we could observe a persistent bias. I think this should be addressed in the discussion, and mentioned somewhere in the introduction. An assessment could be done. For example, is there an actual relationship between the value of the stationarity criteria, and the actual EC roughness values? Also, the presence of turbulent transport can be assessed through a spectral analysis or the analysis of the integral turbulence values.

Katabatic winds

A katabatic wind maximum is often present near the surface above glaciers, and this is mentioned in the text and discussion, but only briefly. Actually the formulation of the bulk method is not adapted to the presence of a katabatic maximum, since it induces turbulent transport (Smeets and al., 1998,2000). Though no real assessment is made upon that.

Assessment of errors and specifically surface temperature.

Errors on measurements on the EC derived roughness are not assessed. Though, these might be large. Also, the stability corrections, which are used to assess the neutrality of the turbulent flow, and to finally calculate the surface roughness, are dependent upon the surface temperature measurements which, I suppose (not stated

clearly in the manuscript) are derived from the Infrared radiometer readings (either an Apogee SI-111 or maybe the Kipp & Zonen CNR4 when the previous is not available). These measurements are directly linked to the value of the surface emissivity of the snow or ice. Which value is assumed for that is not clearly mentioned. How are you taking the uncertainty related to that into account?

A few specific comments.

The previous reviewer already provided the relevant comments upon the part describing the profile and block methods, here are some specific comments on the other parts:

1) Introduction. For roughness assessment you could also use detailed wind-temperature profiles (Sicart et al, 2014) 2.4) Data treatment, eddy covariance data: Could you provide the chosen threshold for stationarity and indicate the percentage of remaining data blocks after filtering? 2.5) mention somewhere how you derive surface temperature out of the SI-112 apogee instruments or the CNR4, more specifically what do you choose for the surface emissivity? 3.1) Line 23: provide the actual range explicitly. 4.1) Line 25: do you have any estimate of the actual height of the Katabatic wind maximum?

References. Barthlott C, Drobinski P, Fesquet C, Dubos T, Pietras C (2007) Long-term study of coherent structures in the atmospheric surface layer. Boundary-Layer Meteorol 125(1):1–24

Högström U, Hunt JCR, Smedman AS (2002) Theory and measurements for turbulence spectra and variances in the atmospheric neutral surface layer. Boundary-Layer Meteorol 103(1):101–124

Litt, M., Sicart, J.-E., Helgason, W., and Wagnon, P.: Turbulence characteristics in the atmospheric surface layer for different wind regimes over the tropical Zongo glacier (Bolivia, 16_ S), Bound.-Lay. Meteorol., 154, 471–495, doi:10.1007/s10546-014-9975-6, 2015.

Smeets CJPP, Duynkerke P, Vugts H (1998) Turbulence characteristics of the stable boundary layer over a mid-latitude glacier. Part I: a combination of katabatic and large-scale forcing. Boundary-Layer Meteorol 87(1):117–145

Smeets CJPP, Duynkerke P, Vugts H (2000) Turbulence characteristics of the stable boundary layer over a mid-latitude glacier. Part II: pure katabatic forcing conditions. Boundary-Layer Meteorol 97(1):73–107

Sicart JE, Litt M, HelgasonW, Ben Tahar V, Chaperon T (2014) A study of the atmospheric surface layer and roughness lengths of the high-altitude tropical Zongo Glacier, Bolivia. J Geophys Res 119(7):3793–3808

Thomas C, Foken T (2005) Detection of long-term coherent exchange over spruce forest usingwavelet analysis. Theor Appl Climatol 80(2–4):91–104

---

## Author Comment (AC1) · 20 Feb 2019

**Response to Reviewers of 'A multi-season investigation of glacier surface roughness lengths through *in situ* and remote observation.'**

Firstly, we wish to sincerely thank the reviewers for providing detailed and thorough reviews of our manuscript. Your comments were extremely useful in refining the paper, and we have endeavoured to respond adequately to each of your comments below. In terms of the main changes to the manuscript, a more comprehensive account of the uncertainties surrounding turbulence measurements and calculations on glaciers has been added to the introduction, methods and discussion, while the information and findings on the potential affects of katabatic wind maximums and non-stationary turbulence have been expanded. Additional expansion and clarification of the methods employed in this study has also been performed throughout the text. Please find below our responses to the individual comments of the reviewers.

**Response to Reviewer 1**

**1) Other Studies Estimating Roughness on Mountain Glaciers:**

*The introduction is clear and concise, presenting a clear rationale for the study while acknowledging previous work. In general, this paper adequately cites and gives ample credit to other studies. However, I can identify a few specific papers that contain findings of some relevance to the study here. In the majority, they add greater weight to the arguments and findings presented. While I agree with the comment (P3, L6) that "Similar studies on mountain glaciers are very rare", there are a few examples for Himalayan (debris-covered) glaciers that might be worth considering (Quincey et al., 2017; Miles et al., 2017).*

The suggested studies have now been referenced in the manuscript, with an outline of their findings presented in the introduction.

**2) Block Estimation Method:**

*Equation (7) would benefit from some clarification. It took me a while to understand that, but I was simply confused by the definition of 'lines' and 'rows'. Aside from that, it makes sense – Figure 2 is extremely helpful. How does this work when the blocks have a larger cell dimension? Do you assume sheltering across the whole 11 m?*

We have added additional text here, referencing Fig. 2, to help clarify the values going into Eq. 7. The same process is applied for all ranges of block sizes, and as a result, the extent of the assumed sheltering will increase with increasing block size. Therefore, by varying the size of the block, we were also testing different assumed sheltering ranges. We have added a line to the methods in Section 2.6.1 to clarify this, and it is discussed in Section 4.2.

*How were edge effects of using a moving window dealt with? Was it extrapolated or padded with zeros?*
The main step taken to avoid edge effects influencing the calculated roughness values was to use large subareas of the DEMs (2000 x 2000 m) around the stations. Considering that the vast majority of the turbulent footprint (and associated weighting) was contained within 200 m of the station, the values of the grid cells on the edges of the subareas will essentially have no influence on the calculated roughness length values. For populating the values of the edge grid cells, the border around the grid cell of interest (i.e. the block size) was reduced to fit the available grid cells. For example, when using the 3 x 3 m block size, a 2 x 3 m or 3 x 2 m block size was used for the edge grid cells.

*Is the result of equation 8 not really a localised z0? Why is it given different notation and called a drag value?*
We wished to use different notation for the values produced from Eq.8 as we recognise the momentum roughness length to be a function of the airflow interacting with the surface of the upwind footprint. As the value calculated at Eq.8 is very localised and does not account for the net effect of the turbulent footprint (that comes in the next step), we felt naming it as a momentum roughness length would not be correct. We therefore hold off referring to momentum roughness lengths until we have weighted the values from Eq.8 over the turbulent footprint (Eq.9).

*What are these "assumed footprint areas" (P9L18)? It sounds interesting, but I'd like to see more details (mentioned again P18, L7). Actually, the finding on P12L30 that equally weighted cells gave similar results is very interesting and helpful for studies going forwards. Is the assumed footprint related to this?*

Yes, this finding is referring to the assumed footprint. A series of assumed footprint areas were tested, ranging from 51 x 51 m to 251 x 251 m in size, and located directly upwind of the station grid cell. The drag values for each cell within these areas were weighted equally when calculating the momentum roughness length value. This gave a roughness length value for an assumed footprint area without the use of EC-derived footprint data, and as you have stated, this was done to see if future studies without EC data could apply this assumption. Implementing a 101 x 101 m with equally weighted cells returned values in line with the EC-derived values. We have added additional text to the methods and discussion section to clarify this.

**3) Profile Estimation Method:**
*I found this method to be very interesting. In fact, it differs markedly from the conventional 'profile' approach (e.g. Munro, 1989) in that wind parallel profiles are taken, rather than perpendicular profiles. This works well when dealing with topographic data at the scale available here (1 x 1 m). I find myself agreeing with the authors' approach here, but I think the difference between this and previous work could be flagged.*
Additional text has been added to Section 2.6.2 to highlight the similarities and differences in this method with previous techniques. Fundamentally, this method is based on the theory of Lettau (1969), as is the case with the technique used by Munro (1989). As noted, however, it differs in its application of this theory by examining wind-parallel rather than wind-perpendicular profiles.

*The cut-off wavelength of 35 m seems to be important, but looking at Figure 3c, I
cannot quite see why this was decided upon.*

We assumed that it would be the smaller wavelength features on the surface (e.g. melt channels, crevasses, boulders) that would disturb air flow and influence the roughness length values, while air flow would likely follow the larger wavelength features (e.g. surface undulations due to bed topography). Therefore, we wanted to separate the smaller scale features from the larger scale features at the test sites. Without defined criteria to make this selection, we determined a cut-off wavelength by analysing the power spectrum of the detrended profiles at each location. We looked for a band of wavelengths, with spectrum at zero, that was located between the energy from small and large wavelengths i.e. we looked for a separation of scales based on the power spectrum. At each site, a similar wavelength of approx. 35 m was identified, and we chose this value as the maximum wavelength influencing roughness length, and filtered out larger wavelength features from the roughness calculations. We hypothesise that 35 m appears to perform well as the cut-off wavelength for these calculations as it is likely similar to the height of the stable boundary layer over the glacier surface, and indicates the max wavelength of the features that this shallow air flow would be impeded by rather than follow. Text has been added to Section 2.6.2 and 4.2 to clarify this.

*Equation 10 shows that the absolute value of the elevation difference between cells
is used in the estimate of s. Is this correct? Is it not more appropriate to consider
the differences facing the wind (i.e. only considering positive differences, rather than
turning negative differences into positive values)?*

The layout of this section has been edited and additional text and equations (Eq. 10 and 11 in the revised manuscript) have been added to clarify the steps taken here, and to facilitate reproducibility. Only the positive differences in surface height are considered. Division by 2 in the calculation of $s$ (Eq. 12 in revised manuscript) is employed so that only the absolute height deviations above the mean height are accounted for in the roughness calculation. The distribution of the height values $h$ around the mean was close to symmetrical, i.e. the mean was equal to the median, and therefore the division by 2 is appropriate.

*As with the block method, how were edge effects dealt with?*

The detrending and filtering in the profile method was applied over profiles 600 m in length (300 m upstream and downstream of the grid cell of interest), while the calculation of roughness lengths was applied on a fetch up to 70 m upstream of the grid cell of interest. Therefore, the edge effects of detrending and filtering did not impact the domain used in the roughness length calculations.

**4) DEM Scale Sensitivity:**

*I find the creation of the synthetic DEM at a finer scale than the LiDAR data to be the
least convincing aspect of the paper. In my opinion, it detracts from the key messages
of the paper and is better off being removed entirely. It would make a useful discussion
point for further work, but I am not convinced that the data support this analysis.*

Ideally, the DEM-based roughness methods would be tested on a range of DEM resolutions, and this recommendation has been further emphasised in the conclusions. In the absence of these data sources, however, we believe it is important to provide some form of sensitivity analysis in the paper to highlight the uncertainties in these methods before they may be employed in another study. Therefore, we suggest that retaining this sensitivity test strengthens the analysis presented in this paper, as a whole.

Technical corrections:

*P4L22: data was -> data were*
Corrected

*Table 2 – are z and zu defined in the text anywhere?*
Text added to Table 2 caption to clarify, and pre-exists in Section 2.5.

*P10L30 – should the second instance of 'parallel' read 'perpendicular'?*
Yes, corrected.

*P10L33 –re-word 'demeaned'!*
Corrected

*Section 3.2.3 – to confirm, is this the 1 x 1 m data?*
Yes. Text added to confirm this

*P16L32 – note typo in citation*
Corrected

**Response to Reviewer 2**

*This paper should really be published, since the work is highly relevant for future investigation. Though, a few important key issues, pointing towards the basis of the above mentioned methods, are only briefly mentioned, thought they might play a fundamental role in the divergences observed in the results from the different methods. I don't think any key calculations would have to be remade, but at least a stronger discussion of these issues should be included, and an assessment of the impact that may have on the observed results, in order for readers and the community not to blindly follow these.*

To strengthen the discussion and to highlight potential sources and impacts of uncertainty and error in the methods and calculations in this study, a number of additions have been made to the manuscript. Further consideration has been given to the uncertainties surrounding turbulence measurements and calculations on glaciers in the introduction, methods and discussion. The presence and height of a katabatic wind maximum at the study sites and the potential impact on roughness calculation has been discussed. In addition, details on the test for non-stationary turbulence have been expanded, and a determination of the random error in the eddy covariance measurements has been added. Further details are provided below.

*The eddy covariance derived surface roughness and the assumptions of the similarity theory. All along the paper, it is assumed that the 30 min-averaged values of u\*, T\* and q\*, are collected during flow conditions which are representative of the assumptions of the similarity theory, a necessary condition for the applicability of the bulk-formulation. This is guaranteed through the careful filtering and selection of the high frequency data, for various criteria (i.e, minimum wind speed, specific flow direction, near-neutral stability conditions, etc., section 2.4, lines 20 to 28). This ensures the scaling of the bulk formulation is valid only if the state of the turbulent flow is driven only by its interaction with the surface, in other words if turbulence is generated only by the interaction of the flow with the surface, and depends only on the stability of the surface layer. One key filter that ensures this conditions to be fulfilled is the one for stationarity of the flow and fully developed turbulence. Thought, even in these cases it is possible that additional turbulence is transported through by large eddies originating from outside*

*layers – related processes, or being transported away, so that the actual measured u\*, T\*, q\* do not scale with the mean flow surface properties (Hogstrom, 2002, Thomas and Foken 2005, Barthlott et al 2007, Litt et al., 2015), even under neutral stability conditions. In such cases it is most likely that the roughness provided by the High Frequency fluxes will not be representative of the surface characteristics, and so that the roughness inferred out from the actual surface state (DEM block, profile methods) do not reproduce the actual fluxes when using the bulk formulation. If the filtering retains only specific meteorological conditions for which the turbulent flow develops in a certain way, we could observe a persistent bias. I think this should be addressed in the discussion, and mentioned somewhere in the introduction. An assessment could be done. For example, is there an actual relationship between the value of the stationarity criteria, and the actual EC roughness values? Also, the presence of turbulent transport can be assessed through a spectral analysis or the analysis of the integral turbulence values.*

A new section has been added to the Introduction to highlight the uncertainties in calculating EC-derived roughness length values through the bulk method. This includes the uncertainty in the applied stability functions, the prevalence of katabatic low-level wind maximums over sloped glaciers and the associated assumptions of the bulk method that may not be valid in such conditions (e.g. constant flux/momentum layer, stationary turbulence, negligible advection etc.). As noted, a substantial series of filters have been applied to the EC data in an effort to obtain roughness values representative of surface interactions. The details on the applied stationarity filter have been expanded in Section 2.5. In addition, a sensitivity test to the selection of the stationarity criteria has been added.

*A katabatic wind maximum is often present near the surface above glaciers, and this is mentioned in the text and discussion, but only briefly. Actually the formulation of the bulk method is not adapted to the presence of a katabatic maximum, since it induces turbulent transport (Smeets and al., 1998,2000). Though no real assessment is made upon that.*

The development of katabatic wind maximums and the associated uncertainties has been further discussed in the introduction (as mentioned in the response above). In addition, a discussion has been added to Section 4.1.1 which indicates that the development of low-level wind maximums near measurement height was likely frequent at the glacier sites (based on observations from a companion study). The majority of such periods, however, would have been identified as being stable and filtered out of the roughness calculations.

*Errors on measurements on the EC derived roughness are not assessed. Though, these might be large. Also, the stability corrections, which are used to assess the neutrality of the turbulent flow, and to finally calculate the surface roughness, are dependent upon the surface temperature measurements which, I suppose (not stated clearly in the manuscript) are derived from the Infrared radiometer readings (either an Apogee SI-111 or maybe the Kipp & Zonen CNR4 when the previous is not available). These measurements are directly linked to the value of the surface emissivity of the snow or ice. Which value is assumed for that is not clearly mentioned. How are you taking the uncertainty related to that into account?*

Text has been added to Section 2.2 to clarify the source of the surface temperature data (observed in 2015 and 2016 by infrared sensor, and estimated from outgoing longwave in 2014) and to complement the existing information in Table 2. The method used to determine surface temperature from outgoing longwave and the choice of emissivity for the 2014 dataset is detailed in the referenced paper (Fitzpatrick et al., 2017). An emissivity value of 0.98 was used for the surface. Varying the emissivity value between 0.95 and 1 (range based on previous snow/ice studies) had negligible impact on the calculated values. The random error values on the EC-observed fluxes have been calculated and added to Section 2.4.1. Furthermore, the standard deviation of the 30-minute roughness lengths from which the seasonal values are calculated are presented in Table 4 to indicate the large range in the EC-observed roughness values, despite extensive filtering. This substantial variability and scattering is noted and discussed in Sections 3.1 and 4.1.1.

Specific:
*1) Introduction. For roughness assessment you could also use detailed wind temperature profiles (Sicart et al, 2014)*
Texted added to introduction

*2.4) Data treatment, eddy covariance data: Could you provide the chosen threshold for stationarity and indicate the percentage of remaining data blocks after filtering?*
Details on the stationarity filter have been added to Section 2.4, and the outcome of this filtering added to Section 3.1.

*2.5) mention somewhere how you derive surface temperature out of the SI-112 apogee instruments or the CNR4, more specifically what do you choose for the surface emissivity?*
As outlined above, text has been added to clarify this.

*3.1) Line 23: provide the actual range explicitly.*
Text added

*4.1) Line 25: do you have any estimate of the actual height of the Katabatic wind maximum?*
The text here in Section 4.1.1 has been expanded, as mentioned above in a previous response, including an indication of wind maximum height.